# C-terminally phosphorylated p27 activates self-renewal driver genes to program cancer stem cell expansion, mammary hyperplasia and cancer

Seyedeh Fatemeh Razavipour [1,2], Hyunho Yoon[2,3], Kibeom Jang[2], Minsoon Kim[2], Hend M. Nawara [1], Amir Bagheri[1], Wei-Chi Huang[1], Miyoung Shin [2,4], Dekuang Zhao[2], Zhiqun Zhou[2], Derek Van Boven[5], Karoline Briegel [2,6], Lluis Morey [2,5], Tan A. Ince[7], Michael Johnson[1] & Joyce M. Slingerland [1,2] ✉

In many cancers, a stem-like cell subpopulation mediates tumor initiation, dissemination and drug resistance. Here, we report that cancer stem cell (CSC) abundance is transcriptionally regulated by C-terminally phosphorylated p27 (p27pT157pT198). Mechanistically, this arises through p27 co-recruitment with STAT3/CBP to gene regulators of CSC self-renewal including *MYC*, the Notch ligand *JAG1*, and *ANGPTL4*. p27pTpT/STAT3 also recruits a SIN3A/HDAC1 complex to co-repress the Pyk2 inhibitor, *PTPN12*. Pyk2, in turn, activates STAT3, creating a feed-forward loop increasing stem-like properties in vitro and tumor-initiating stem cells in vivo. The p27-activated gene profile is over-represented in STAT3 activated human breast cancers. Furthermore, mammary transgenic expression of phosphomimetic, cyclin-CDK-binding defective p27 (p27CK-DD) increases mammary duct branching morphogenesis, yielding hyperplasia and microinvasive cancers that can metastasize to liver, further supporting a role for p27pTpT in CSC expansion. Thus, p27pTpT interacts with STAT3, driving transcriptional programs governing stem cell expansion or maintenance in normal and cancer tissues.

As in normal development, cancers are maintained by stem-like cell populations termed cancer stem cells (CSC), that seed new growth[1]. CSCs are defined by their tumor-initiating ability and can self-renew, generate more differentiated progeny, and mediate therapy resistance and metastasis[1]. In breast cancer, populations with CD44+CD24-/low surface markers and high ALDH1 activity are enriched for cells with stem cell properties[1–3]. These cells rely on expression of stem cell transcription factor genes including *SOX2*, *MYC*, and *NANOG* to generate tumor spheres in vitro and tumors in vivo with high frequency[1–3]. While normal epithelial cells undergo programmed cell death or anoikis upon loss of adhesion to their basement membrane, CSC form spheroids in suspension[4]. During metastasis, cancer cells undergo epithelial–mesenchymal transition (EMT)[5]. CSC enriched populations show a shift to EMT, overexpress drivers of EMT and metastasis, and

[1]Cancer Host Interactions Program, Lombardi Comprehensive Cancer Center, Department of Oncology, Georgetown University, Washington DC, USA. [2]Braman Family Breast Cancer Institute, Sylvester Comprehensive Cancer Center, University of Miami, Miller School of Medicine, Miami, Fl, USA. [3]Department of Medical and Biological Sciences, The Catholic University of Korea, Bucheon-si, South Korea. [4]Department of Pathology, Yale School of Medicine, New Haven, CT, USA. [5]John P. Hussman Institute for Human Genomics, Dr. John T. Macdonald Foundation Department of Human Genetics, University of Miami, Miller School of Medicine, Miami, FL, USA. [6]Department of Surgery, University of Miami, Miller School of Medicine, Miami, Fl, USA. [7]Department of Pathology and Laboratory Medicine, Weill Cornell Medicine, New York, NY, USA. ✉e-mail: js4915@georgetown.edu

are thought to seed metastasis[6–8]. Triple-negative breast cancers (TNBC) lack expression of hormone receptors and *HER2* amplification. While they account for only 15% of breast cancers, they have a very high mortality, affecting younger women, with early recurrences leading to early patient demise[9]. The aggressiveness of TNBC has been attributed to an enrichment of intrinsically therapy-resistant cancer stem cells, leading to metastasis[10,11].

CSC-enriched populations in several different cancer types show constitutive activation of signal transducer and activator of transcription 3 (STAT3)[12]. STAT3 is a critical regulator of normal embryonic stem cell self-renewal[13–15] and has a pivotal role in maintaining CSC self-renewal[10]. STAT3 is frequently constitutively activated in TNBC, contributing to its highly aggressive phenotype[10]. STAT3 plays critical roles in tumor initiation, progression, and metastasis by regulating expression of downstream target genes[16–18]. This makes the STAT3 pathway an attractive pharmacological target in oncology.

p27 is a key cell cycle regulator that plays a dual role in tumorigenesis. p27 acts as a tumor suppressor to inhibit cyclin-CDKs and restrain the cell cycle. Upon C-terminal phosphorylation at T157 and T198 by PI3K activated kinases[19–22], p27 acquires pro-oncogenic actions to promote cancer invasion and metastasis[23–25]. While the mechanisms for this are not clear, p27 also governs differentiation in many tissue types[26–30]. Considerable evidence from murine and *xenopus* developmental studies indicate that at least some of the actions of p27 in development are cyclin-CDK independent, since developmental defects of p27 knock out are restored by knock in of a cyclin CDK-binding defective *CDKN1B* allele (p27CK-)[28,29,31,32]. Notably, knock-in of p27CK- into p27 null mice led to expansion of lung progenitors[23], suggesting a potential role for p27 in tissue stem cell regulation.

C-terminally phosphorylated p27pT157pT198 (p27pTpT) was recently shown to act as a transcriptional co-activator of cJun to drive EMT gene expression and metastasis[33]. p27pTpT also promotes STAT3-mediated EMT and metastasis[34], both of which properties are enriched in CSC populations[6,8]. Since STAT3 is a key driver of embryonic stem cell (ES) and CSC self-renewal[12,14,15], here, we investigate whether p27pTpT can drive stem or progenitor cell expansion in development and in different cancer models through transcriptional coregulation of STAT3. We show that C-terminally phosphorylated p27 is a STAT3/CBP co-regulator governing CSC self-renewal via induction of CSC programs including *MYC, JAG1*, and *ANGPTL4*. p27pTpT/STAT3 also acts as a corepressor with SIN3A/HDAC1 to repress the Pyk2 inhibitor, *PTPN12*. Pyk2 activates STAT3, creating a feed-forward loop to upregulate stem cells in vitro and in vivo. Furthermore, mammary transgenic expression of phosphomimetic, cyclin-CDK-binding defective p27 (p27CK-DD) leads to hyperplasia and microinvasive cancers that can metastasize to liver, further supporting a role for p27pTpT in CSC expansion. Thus, p27 is a master regulator of STAT3-driven CSC transcriptional programs, driving tumor initiation and metastasis in preclinical xenograft and genetic models.

## Results

### p27pTpT upregulates stem cell gene programs and down-regulates differentiation profiles

p27 is known to play a cell cycle independent role as a transcriptional co-regulator of cJun to drive gene programs governing tumor metastasis[33]. CSC represent a subpopulation of bulk tumor cells with greater invasive, metastatic, and self-renewal ability[35,36]. Since prior work indicated a role for p27 in metastasis[33,34,37,38], here we tested if p27 might drive both pro-metastatic and stem cell functions through actions as a transcriptional regulator. Parental MDA-MB-231 (hereafter 231) was compared to a highly bone metastatic sister cell line MDA-MB-231-1833 (hereafter 1833), isolated from bone metastases of xenografted 231[39] that expresses high levels of p27pTpT due to PI3K activation[40]. We also used a 231-derived line expressing a cyclin and CDK-binding defective, C-terminal phosphomimetic p27CK-

T157DT198D (p27CK-DD)[40]. In these cells, lentiviral transduction leads to expression of p27CK-DD levels similar to the increased endogenous p27 in cancer cells with constitutive PI3K activation[41]. Since C-terminal phosphorylation stabilizes p27[42,43] mutant p27 levels were lowest in 231 cells expressing p27CK-AA, increased in 231p27CK- and highest in 231p27CK-DD (hereafter 231DD) see Fig. S1a.

Global p27-regulated gene expression analysis performed earlier[33] was re-analyzed for both up- and down-regulated genes in these triple-negative breast cancer (TNBC) sister cells lines. A profile of p27-upregulated genes (fold change >1.5, q < 0.05) was identified by comparing our 231DD with parental 231. Among the p27 genes upregulated in 231DD versus 231, there were 367 genes that were also down-regulated (fold change <0.75, P < 0.05) upon p27 depletion in 1833shp27 compared to 1833 (Fig. 1a top). In addition, 536 genes were downregulated in 231DD compared to 231 (fold change <0.75, P < 0.05) and upregulated upon loss of p27 in 1833shp27 genes (fold change >1.5, q < 0.05; Fig. 1a bottom).

This analysis showed p27-activated genes associate with pathways governing cancer stem cell self-renewal including ESC pluripotency, Notch, Wnt, and Hedgehog pathways (Fig. 1b top). Notably, the 536 p27-downregulated genes associate with differentiation pathways including ectodermal, oligodendrocyte, and hematopoietic stem cell differentiation pathways (Fig. 1b bottom). Taken together, these data suggest a role for p27pTpT in the activation of stem cell gene programs.

### p27 phosphorylation increases ES-TF expression and sphere forming cell abundance

To further investigate if p27pTpT induces stem cell properties, 1833 and 231DD were compared with 188shp27 and 231, respectively, for tumor sphere formation and expression of embryonic stem cell transcription factors (ES-TFs) both properties of breast CSCs[33]. 231DD shows enhanced expression of ES-TFs *SOX2, MYC*, and *NANOG* (Fig. 1c top) and tumor sphere formation (Fig. 1c bottom) compared to vector control 231 cells. In contrast, 231 cells transduced with p27CK-AA (231AA) show ES-TF levels and tumor spheres similar to control 231 (Fig. 1c). Notably, sphere formation is also increased by p27CK-DD overexpression in the ER + MCF7 breast cancer line (Fig. S1b). We also compared p27 effects in a human bladder cancer model of sister cell lines, including the poorly metastatic parental UMUC3 line and the highly lung metastatic variant, UMUC3-LuL2 line derived from UMUC3[44]. p27CK-DD expression also increased sphere formation in UMUC3, supporting the generalizability of these effects (Fig. S1c). Conversely, p27 depletion in the highly metastatic 1833 breast and UMUC3-LuL2 bladder cancer models (Fig. S1d top and bottom, respectively) significantly reduces ES-TFs expression (Fig. 1d left, S1e) and tumor sphere abundance (Fig. 1d right, S1f).

MCF12A is an immortal, but non-tumorigenic human mammary epithelial line that forms spheres slowly over 4–6 weeks. Transduction of a vector encoding p27CK-DD, but not p27CK-, significantly increases mammosphere formation in MCF12A compared to vector controls (Fig. 1e). Furthermore, p27CK-DD expression in MCF12A causes a shift from predominantly CD44$^{low}$/CD24$^+$ to a greater population expressing stem cell surface markers, CD44$^+$CD24$^{low/-}$ (Fig. 1f left), a hall mark of breast cancer initiating cells or CSCs[2]. CD44 protein levels markedly increase with expression of p27CK-DD but not p27CK- (Fig. 1f, right) MCF12Ap27CK-DD cells but not MCF12Ap27CK- also acquire the ability to form soft agar colonies (Fig. 1g). 231DD also form more soft agar colonies than 231, and colony formation decreases with p27 knock-down in 1833 cells (Fig. 1h).

Key stem cell transcription factors, Nanog, Oct4, and Sox2 (NOS) are highly enriched and essential for self-renewal of embryonic stem cells (ES)[45]. In ESC, these TFs regulate each other and target gene mediators of differentiation[45]. We compared our p27 activated gene

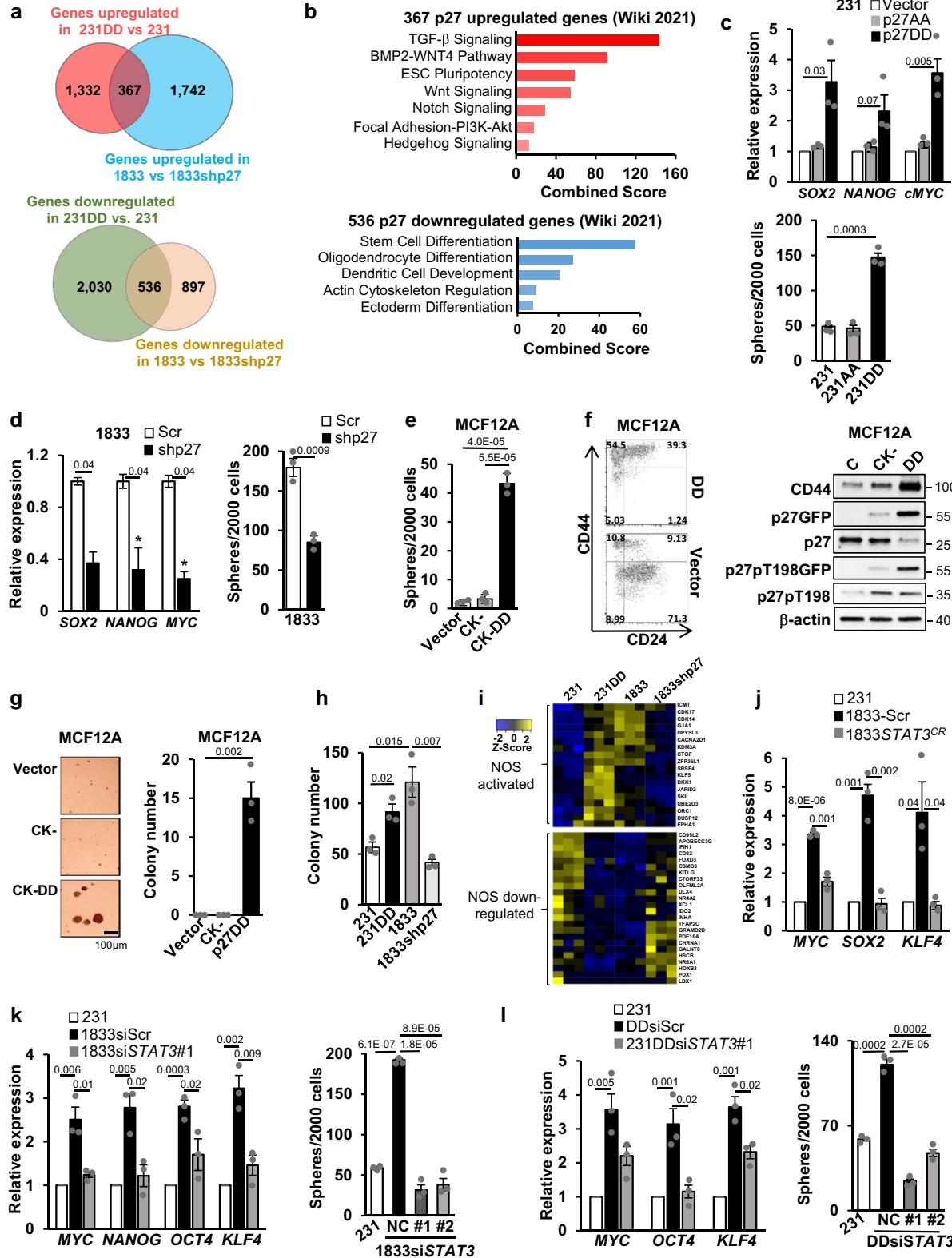

profile with a profile of NOS target genes identified in ES. Notably, a subset of NOS activated genes is coordinately upregulated and NOS repressed genes are downregulated in p27-activated 231DD and 1833 compared to 231 and 1833shp27 (Fig. 1i). Pathway analysis of genes shared in the NOS target signature and in our p27-activated lines identifies important stem cell and developmental pathways including Wnt, pluripotent stem cell and AP-2 transcriptional regulation

pathways (Fig. S1g, h). These data indicate that p27pTpT or p27CK-DD expression increases stem-like cells and stem cell-related gene expression.

**p27 action on stem cell properties is mediated through STAT3**
p27 can bind STAT3, a driver of ES and CSCs[12,14,15], and facilitates its activation[34]. Thus, we investigated if p27 action to increase stem cell

**Fig. 1 | p27pTpT increases stem-like cells through STAT3. a** Venn diagrams show p27-upregulated and repressed genes. **b** Pathway analysis of p27-up (top) and down-regulated (bottom) genes. **c** Effects of p27CK-DD and p27CK-AA transduction in 231 cells on embryonic stem cell transcription factors (ES-TFs) expression (top) and mammosphere formation (bottom). **d** Effect of p27 depletion in 1833 cells on ES-TFs expression (left) and mammosphere formation (right). **e** Effect of p27CK- and p27CK-DD transduction in MCF12A on mammosphere formation. **f** Surface expression of CD44 and CD24 in MCF12A-vector, MCF12A-p27CK-, and MCF12A-p27CK-DD assayed by flow cytometry (top). Western shows CD44, p27, and p27pT198 with β-actin controls (bottom). **g** Effect of p27CK- and p27CK-DD transduction in MCF12A on colony formation. **h** Effect of p27CK-DD transduction in 231 and p27 knockdown in 1833shp27 vs 1833 on colony formation. **i** Gene expression heatmaps of NOS (Nanog, Oct4, and Sox2) target gene signatures in indicated lines. **j–l** Effects of STAT3 depletion on p27-driven increase in ES-TFs expression and sphere formation in indicated cells. All graphs show mean ± SEM from $N = 3$ biological replicate assays. *p*-values were represented by paired one-tailed Student's T Test (*p* values are shown in graphs). Source data are provided as a Source data file. For ES-TFs, all of *SOX2, NANOG, MYC, KLF4*, and *OCT4* were assayed, but only positive data were graphed. ESC embryonic stem cell, C control, NC negative control.

properties requires STAT3. Notably, *STAT3* CRISPR knockout (Fig. S1i), transient *STAT3* knockdown by two different siRNAs (Fig. S1j) and treatment with a STAT3 inhibitor (DPP) (Fig. S1k) all reverse the p27pTpT-mediated ES-TF upregulation in 1833 cells (Figs. 1j, 1k left, S1l, S1m). In addition, while higher p27pTpT increases sphere formation in 1833 cells, treatment with two si*STAT3* (Fig. 1k right), and STAT3 inhibition (Fig. S1n) all impair the p27pTpT-mediated increase in tumor spheres. The p27CK-DD-driven increase in ES-TFs expression was decreased by *STAT3* siRNA knockdown (Fig. 1l left, S1o, p), and STAT3 inhibition (Fig. S1q), as was the gain of sphere formation in 231DD compared to 231 (Fig. 1l right, S1r). These data show that p27pTpT induced stem cell properties are mediated, at least in part, through STAT3. Since p27 can co-regulate cJun[33], and cJun and STAT3 are known to interact[41], we next investigated if p27 might act as a co-activator of STAT3 to promote transcription of gene programs governing CSCs expansion.

## p27 co-occupies chromatin with STAT3 and promotes its binding enrichment

Global chromatin occupancy by STAT3 in 1833 and 1833shp27 in the present analysis was compared with that reported for p27 in these lines[33]. This revealed that p27 co-occupies nearly 45% of STAT3 binding sites in 1833 cells (Fig. 2a left). Furthermore, STAT3 binding to p27/STAT3 co-bound peaks was significantly reduced upon p27 depletion in 1833 cells (Fig. 2a right). Genomic distribution analysis showed that over half of the peaks bound by STAT3, p27, and both STAT3/p27 localize to promoters or intronic sites, with remaining sites occupying intergenic regions (Fig. 2b). The heat map Fig. 2c, middle portion, shows that of 16,697 STAT3 peaks co-bound by p27, 73% of these, or 12,248, were lost or decreased in amplitude with p27 knockdown. An additional set of 4171, exclusively STAT3-bound peaks were acquired in 1833shp27 cells (Fig. 2c right, top) and 4458 STAT3-bound peaks were unaffected by p27 loss (Fig. 2c right, bottom).

Strikingly, 65% of STAT3 target genes (defined as bound +/−5 kb of the transcription start sites or TSS) are co-bound by p27 in 1833 cells (Fig. 2d left). STAT3 binding enrichment was reduced at over 70% of these p27/STAT3 co-bound target gene promoters upon p27 knockdown (Fig. 2e, f right, adjusted *p* value 2.81E−04). Notably, pathway analysis of the p27/STAT3 co-target genes in which STAT3 promoter enrichment was p27-dependent (Figs. 2e, 2f right) revealed their association with stem cell-related Notch and Wnt pathways (Fig. 2g).

## p27/STAT3/cJun co-targets are associated with stem cell pathways

DNA binding motif analysis showed that STAT3 and Jun/AP1 motifs are frequently found within p27 binding sites, STAT3 binding sites and sites co-bound by p27/STAT3/cJun (Fig. 2h). Comparison of STAT3 ChIPseq with p27 and cJun ChIPseq analyses revealed that 95% of the 4001 p27/STAT3 co-target genes identified in Fig. 2d left are also bound by cJun within the region of STAT3/p27 co-binding (Fig. 2i). Thus, p27, STAT3 and cJun are recruited to sites in close proximity within common target genes.

The expression of the p27/STAT3/cJun co-targets was next evaluated. Global expression analysis revealed 196 genes are significantly p27-activated (upregulated by >1.5 fold, q < 0.05) in both 231p27CK-DD and 1833 cells compared to control 231, and down-regulated by p27 knockdown (Fig. 3a, b). Nearly half of the 196 p27-activated genes (*n* = 96), are direct p27/STAT3/cJun targets (Fig. 3c). Notably, p27 depletion in 1833shp27 not only dramatically reduces detectable p27 binding at these target genes (Fig. 3d, top, *p* = 6.96E−49), but importantly, both STAT3 and cJun enrichment at the promoters of these 96 p27/cJun/STAT3 co-targets are significantly reduced by p27 depletion (Fig. 3d, bottom, *p* values 3.19E−13 and 6.89E−19 for STAT3 and cJun, respectively). These 96 p27/STAT3/cJun co-bound genes whose expression is p27-dependent are strongly associated with CSC pathways including Wnt, Notch, and embryonic stem cell pluripotency pathways and mammary development (Fig. 3e). These data support a model in which p27pTpT co-regulates the action of STAT3 and cJun to drive expression of CSC gene programs.

## p27 is co-recruited to chromatin with STAT3, cJun, and CBP to activate stem cell-driver genes

p27, STAT3, and cJun are all recruited to promoters of ES-TF genes that have important roles in CSC self-renewal, including *MYC, SOX2, OCT4*, and *KLF4* (Figs. 3f, S2a−c), Wnt transcription factor (*LEF1*) (Fig. S2d), and the Notch signaling ligand (*JAG1*) (Fig. S2e), and p27 and cJun binding to these promoters is greater in 231DD and 1833 than in 231 controls. Notably, p27 depletion decreases the binding enrichment of both STAT3 and cJun at these p27/STAT3/cJun co-target sites (Figs. 3f and S2a−e).

We next validated p27/STAT3/cJun recruitment to the *MYC* and *JAG1* promoters and p27 effects on their expression by ChIPqPCR and qPCR. 231DD and 1833 both have higher *MYC* and *JAG1* expression levels than in 231 cells (Figs. 1c top, S3a) and p27 knockdown in 1833 decreases expression of both (Figs. 1d left, S3a). 231AA cells showed similar levels of *MYC* and *JAG1* to those in 231 (Figs. 1c top, S3a). Kaplan−Meier analysis showed that higher expression of each of *MYC* and *JAG1* is associated with shorter breast cancer patient relapse-free survival (Figs. 3g, S3b), underlining their oncogenic roles in human breast cancer. ChIPseq showed that p27, STAT3, and cJun can each bind to the *MYC* and *JAG1* promoters and that p27 depletion decreases both STAT3 and cJun recruitment to these sites (Figs. 3f, S2e). ChIPqPCR confirmed that p27, STAT3, and cJun recruitment to these promoter sites was greater in p27-activated 231DD and 1833 than in 231 controls, and was decreased upon p27 depletion and by PI3K inhibitor pre-treatment in 1833 (*MYC* data in Fig. 3h−j, and *JAG1* data in S3c−e). PI3K treatment abrogates p27 C-terminal phosphorylation in 1833[40].

p27pTpT also increases enrichment of the transcriptional co-activator, CBP, at sites of p27/STAT3/cJun occupancy on both the *MYC* and *JAG1* promoters and CBP recruitment in 1833 decreased by p27 depletion and by PI3K inhibition (Figs. 3k, S3f). Higher CBP recruitment is associated with increased H3K27 acetylation at both of these promoters (Figs. 3l, S3g) and with *MYC* and *JAG1* induction in

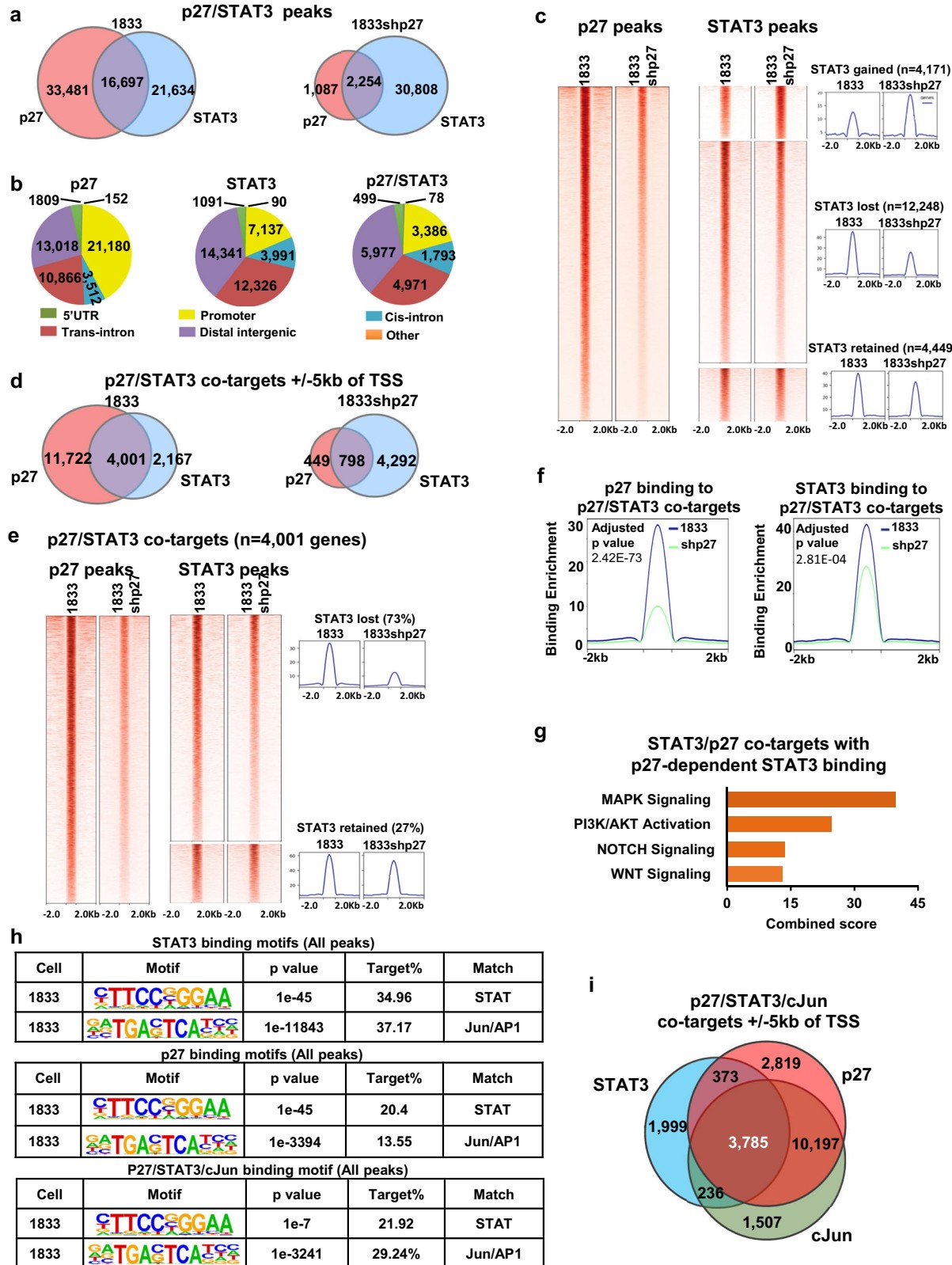

**Fig. 2 | p27 promotes STAT3 binding enrichment on chromatin. a** Venn diagrams of p27 and STAT3 binding peaks in 1833 and 1833shp27 cells. **b** Localization of p27 and STAT3 peaks. **c** p27 and STAT3 DNA binding heatmaps show effects of p27 depletion on STAT3 binding in 1833 and 1833shp27 cells. **d** Venn diagrams show how p27 knockdown affects numbers of target genes bound by STAT3 and/or p27 at promoter sites ±5 kb from TSS in 1833 and 1833shp27 cells. **e** DNA binding heatmaps for p27 and STAT3 binding +/−5 kb from Transcription Start Site (TSS) of 4001 co-target genes in 1833 and 1833shp27 cells. **f** Enrichment of each of p27 and STAT3 at the 4001 p27/STAT3 co-target genes (+/−5 kb TSS) is reduced by p27 knockdown. **g** Pathway analysis of STAT3/p27 co-target genes that have p27-dependent STAT3 binding to their promoters. **h** Top binding motifs at sites of STAT3 binding and for regions bound by STAT3, p27, and cJun. **i** Venn diagram shows numbers of target genes bound by STAT3, cJun, and p27 at +/−5 kb of their TSS. *p*-values were represented by unpaired two-tailed T Test. Source data are provided as a Source data file.

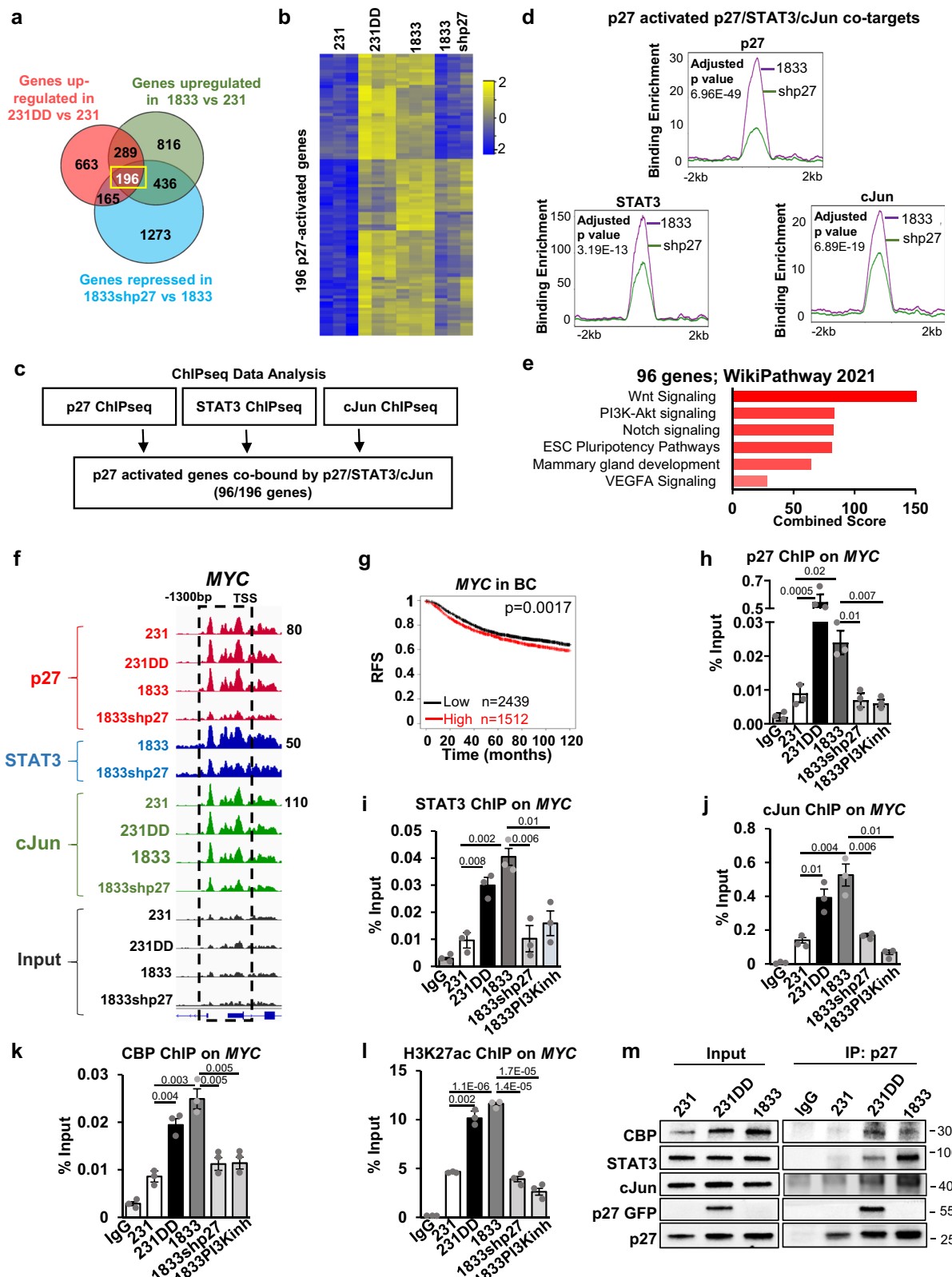

p27-activated lines (Figs. 1c top, 1d left, and S3a). These data support a model whereby p27 enrichment at STAT3, and/or cJun-bound *MYC* and *JAG1* promoters nucleates factor retention to support induction of these co-target genes. Co-IP showed that cellular p27 associates with STAT3, cJun, and CBP in 231 parental controls cells and this is greatly increased in p27CK-DD transduced cells and also in 1833 cells with high endogenous p27pTpT (Fig. 3m).

## p27/STAT3/cJun increases stem cell abundance in part through *ANGPTL4*

*ANGPTL4* confers anoikis resistance, a property that supports survival of tumor cells when they lose adhesion during the metastatic process[46,47]. We identified *ANGPTL4* as a p27-activated target gene: p27CK-DD but not p27CK-AA transduction upregulates *ANGPTL4* expression in 231DD compared to 231, and p27 depletion decreases it

**Fig. 3 | p27 is co-recruited to chromatin with STAT3, cJun, and CBP to induce stem cell-driver gene expression. a** Venn diagram shows p27-upregulated genes. **b** Gene expression heatmaps of p27-upregulated genes. **c** Schematic of p27-activated genes with p27/STAT3/cJun colocalization at their promoter. **d** p27, STAT3 and cJun binding enrichment at the promoters (+/−5 kb of Transcription Start Site (TSS)) of 96 p27/STAT3/cJun bound p27-activated genes in 1833 and after p27 depletion. *p*-values were represented by unpaired two-tailed T Test. **e** Pathway analysis of 96 p27/STAT3/cJun co-targets that are p27-upregulated. **f** p27,

STAT3, and cJun co-occupied peaks at *MYC* promoter. **g** KM plot shows reduced relapse-free breast cancer survival with increased *MYC* expression *p* = 0.0017, *n* = 3951 cancers. **h**–**l** ChIP-qPCR shows p27 (**h**); STAT3 (**i**); cJun (**j**) and CBP (**k**) binding, and H3K27 acetylation (**l**) at the *MYC* promoter. **m** Immunoprecipitation (IP) shows p27CK-DD-dependent p27, STAT3, and CBP association. All graphs show mean ± SEM from *N* = 3 biological replicate assays. *p*-values were represented by paired one-tailed Student's T Test. *p* values are shown in graphs. Source data are provided as a Source data file. ESC embryonic stem cell.

in 1833 (Fig. 4a). STAT3, cJun, and p27 binding sites localized to −2 kb of the *ANGPTL4* TSS (Fig. 4b). STAT3, cJun, CBP, and p27 co-recruitment to the *ANGPTL4* promoter was confirmed by ChIP-qPCR and is greater in 231DD and 1833 than in 231 (Fig. 4c–f). p27 depletion and PI3K inhibition in 1833 both significantly attenuate STAT3, cJun, and CBP recruitment to this site (Fig. 4c–f). p27pTpT might facilitate stable association of STAT3/cJun/CBP to this promoter site or enhance complex stability. Enrichment of CBP on the *ANGPTL4* promoter of 231DD and 1833 cells (Fig. 4f) was associated with higher H3K27ac, and both decrease upon p27 depletion and PI3K inhibition in 1833 (Fig. 5f, g). The biological importance of *ANGPTL4* upregulation is supported by Kaplan−Meier analysis that shows shorter time to disease recurrence in TNBC patients whose tumors have increased *ANGPTL4* expression (Fig. 4h). To test if p27 mediates CSC expansion through Angptl4, *ANGPTL4* expression was depleted by each of two *siANGPTL4* in highly metastatic 1833 and in 231DD (Fig. S3h, i). In both lines, this reduced the p27-mediated upregulation of ES-TFs and decreased tumor spheres (Figs. 4i–l, S3j, k). Thus, p27 driven CSC expansion is mediated at least in part through induction of *ANGPTL4*, a p27/STAT3/cJun co-target gene.

## p27 represses *PTPN12* and increases phospho-Pyk2 to activate STAT3

We next evaluated the 536 genes that are downregulated upon p27 C-terminal phosphorylation (Fig. 1a bottom). *PTPN12* is one of the genes most strongly downregulated in 231DD and 1833 compared to 231 control cells (log2 fold change = −2.82, *p* = 7.05e−293, q = 1.15e−289) (Fig. 5a, b). Notably, levels in 231AA are similar to those in 231. Decreased *PTPN12* expression was also observed in p27CK-DD expressing MCF12A vs vector control (Fig. S4a, b). PTPN12 dephosphorylates mediators of focal adhesion signaling including Pyk2, Paxillin, and p130$^{Cas\,49}$. Pyk2 is a FAK family member tyrosine kinase that promotes EMT, invasion, and metastasis in human breast and other cancers[49]. *PTPN12* depletion with each of two siRNA increased levels of active phosphorylated Pyk2, and of 2 other substrates Paxillin, p130$^{Cas}$ in 231 cells (Fig. S4c).

Pyk2 binds STAT3 to mediate its activating phosphorylation at Y705[50]. Here, we find p27CK-DD increases activated phospho- Pyk2 and STAT3 in 231 (Fig. 5c). Pyk2 inhibition by PF431396 (Fig. S4d top) decreases both pPyk2 and STAT3pY705 in 1833 (Fig. 5d) and in 231DD cells (Fig. S4e). Both stable (Fig. 5c) and acute *PYK2* depletion (Fig. 5d, S4d bottom) also reduce p27pTpT-driven STAT3 activation but not total STAT3 levels indicating that pPyk2 is required for STAT3 activation in these cells. Loss of C-terminal phosphorylation of p27 upon PI3K inhibition (PF1502) abrogates both Pyk2 and STAT3 activation, without affecting total kinase levels (Fig. 5d), supporting a requirement for phosphorylated p27 in Pyk2-dependent STAT3 activation.

Since *PTPN12* expression is downregulated in p27CK-DD and 1833 compared to 231, we next assayed if p27 interacts with STAT3/cJun to repress *PTPN12*. ChIPseq identified STAT3, cJun, and p27 binding at +2 kb of TSS on *PTPN12*, all of which decreased upon p27 depletion in 1833 (Fig. 5e). ChIPqPCR confirmed that p27, STAT3, and cJun are co-recruited to this +2 kb *PTPN12* site, with greater binding in 231DD and 1833 than 231. Notably, in 1833, p27 depletion and pre-

treatment with PI3K inhibitor, PF1502 attenuated p27, STAT3, and cJun recruitment to this *PTPN12* regulatory site (Fig. 5f–h), indicating that STAT3/cJun co-recruitment or retention at this site are p27 dependent. These findings were validated in human bladder cancer sister cell lines[44]. As for the breast cancer lines, cJun and p27 were recruited to the +2 kb *PTPN12* site in a p27-dependent manner, with higher enrichment in UMUC3 overexpressing p27CK-DD than in vector control and loss of cJun binding upon p27 knockdown UMUC3-LuL2 (Fig. S4f, g).

To further investigate STAT3/cJun/p27-mediated gene repression, we assayed the involvement of co-repressors, YY1 and SIN3A, that recruit HDAC1 to other developmentally important transcription factors[51]. SIN3A, YY1, and HDAC1 were detected at the *PTPN12* TSS and their recruitment was greater in p27CK-DD expressing 231 and 1833 than in 231, and reduced by p27 depletion and by PI3K inhibition in 1833 (Fig. 5i-k). Moreover, cJun ChIP-Western also showed SIN3A, YY1, and HDAC1 enrichment at this site was p27-regulated (Fig. 5l). p27 can form a transcriptional repressive complex with E2F4[52,53]. However, ChIPqPCR with anti-E2F4 showed no E2F4 recruitment to the STAT3/cJun/p27 binding site at +2 kb on *PTPN12* TSS (Fig. S4h). Reduced *PTPN12* expression correlates with improved breast cancer patient outcome on Kaplan−Meier analysis (*p* = 0.00651), supporting a tumor suppressor role for PTPN12 in human breast cancer (Fig. S4i). Our findings indicate that p27 is recruited to sites of STAT3, cJun, and CBP binding on stem cell-related gene promoters, potentially stabilizing co-regulator interaction, to induce gene expression (Fig. 5m, left). In addition, activated p27, STAT3, and cJun can also recruit SIN3A, YY1, and HDAC1 to repress *PTPN12* (Fig. 5m right). This would create a feed forward loop to increase Pyk2 activity and further activate STAT3 to drive CSC expansion.

## p27 increases stem-like cells in vitro through loss of *PTPN12* and activation of Pyk2

To address the role of PTPN12 in p27-regulated tumor initiation, *PTPN12* was stably depleted in 231 and UMUC3, and overexpressed in 1833 (Fig. S5a–c). *PTPN12* depletion (sh*PTPN12*) in the weakly metastatic breast (231) and bladder cancer lines (UMUC3) increased ES-TF expression (Figs. 6a, S5d) whereas *PTPN12* transduction in 1833 decreased *NANOG, MYC*, and *SOX2* expression (Fig. 6b). *PTPN12* depleted 231 and UMUC3 formed more spheres (Figs. 6c, S5e) and *PTPN12* overexpressing 1833 formed fewer (Fig. 6c). Effects on soft agar colony formation were similar: *PTPN12* depletion increased and *PTPN12* overexpression decreased colony formation (Fig. 6d).

Stable *PTPN12* depletion activated pPyk2 in 231, while *PTPN12* overexpression decreased pPyk2 in 1833 (Fig. S5a, c). Notably, in 231DD, Pyk2 depletion reversed the stem cell effects of p27CK-DD, decreasing ES-TFs expression (Figs. 6e, S5f), sphere formation (Fig. 6f) and ALDH1 activity (Fig. 6g). Treatment with Pyk2 inhibitor, PF431396, also attenuated sphere formation in 1833 cells (Fig. 6h). Notably, PI3K inhibition over 48 hrs decreased AKTpT308, and led to loss of p27pT198, and pPYK2, and reduced MYC and SOX2 levels (Fig. S5g), consistent with a model in which PI3K drives p27 phosphorylation leading to *PTPN12* repression, Pyk2 and STAT3 activation and CSC expansion.

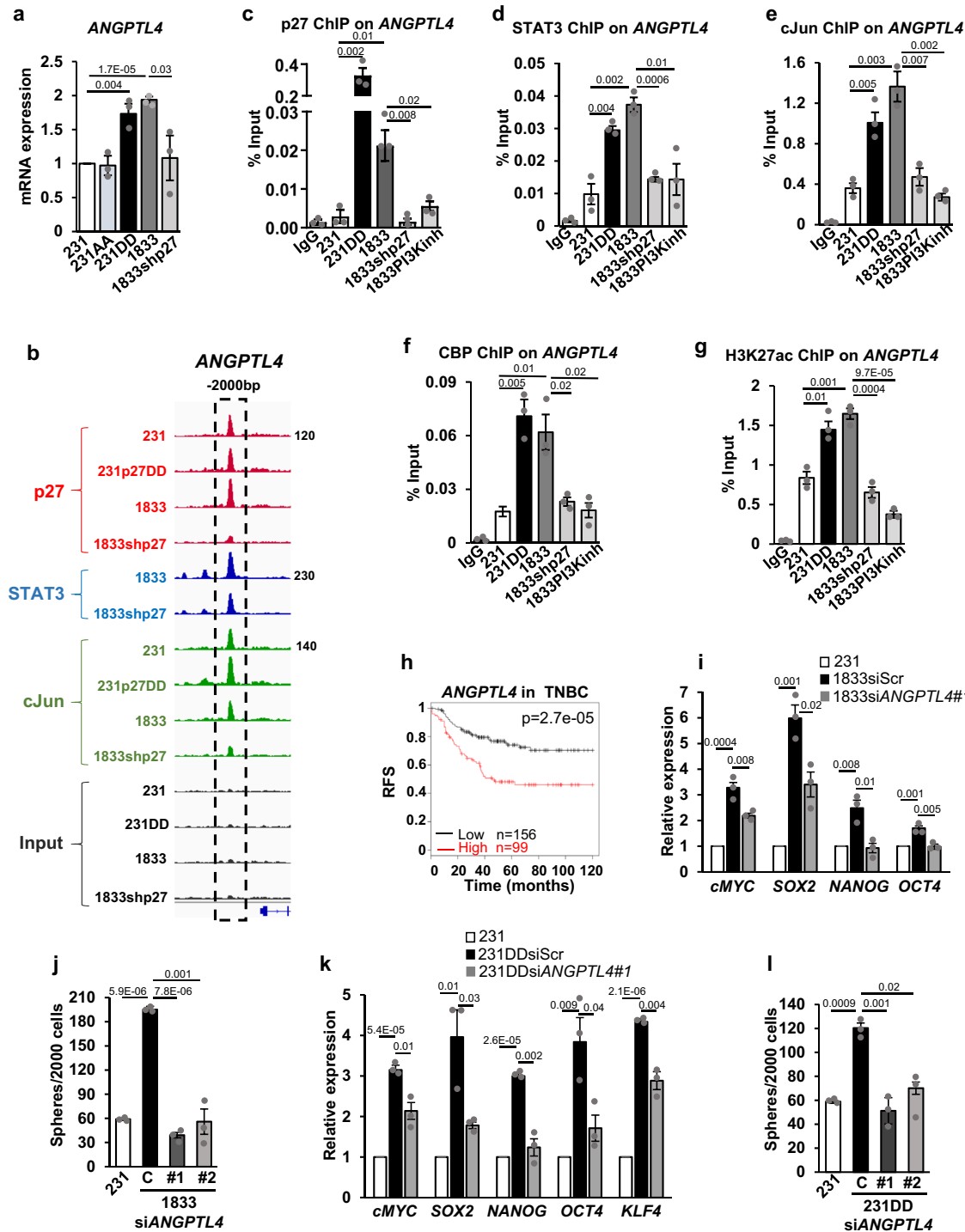

**Fig. 4 | p27 drives stem cell properties in part through induction of *ANGPTL4* expression. a** *ANGPTL4* mRNA expression. **b** ATAC-seq, p27, STAT3, and cJun co-occupy peaks at −2 kb of the *ANGPTL4* TSS. **c**–**g** ChIPqPCR depicts p27(**c**); STAT3 (**d**); cJun (**e**), and CBP (**f**) binding and H3K27 acetylation (**g**) at −2 kb of the *ANGPTL4* TSS. **h** KM plot shows decreased relapse-free triple-negative breast cancer (TNBC) survival with increased *ANGPTL4* expression. **i**–**l** Effects of ANGPTL4 depletion on p27-driven embryonic stem cell transcription factors (ES-TFs) expression and mammosphere formation in indicated cells. All graphs show mean ± SEM from *N* = 3 biological replicate assays. *p*-values were calculated by paired one-tailed Student's T Test and are shown in graphs. Source data are provided as a Source data file.

## Upregulation of tumor-initiating stem cells by p27 in vivo requires cJun, PTPN12 loss and Pyk2

Since p27/STAT3/cJun complexes activate gene drivers of stem cell expansion and repress genes associated with differentiation (Figs. 1 and 2), we next assayed effects of p27 on tumor-initiating ability in vivo. Tumor-initiating stem cell (T-ISC) abundance was assayed in vivo by injecting limiting dilutions of 231, 231DD, 1833, and 1833shp27 (10, 100, or 1000 cells/mouse) orthotopically into Balb/c nude mice. T-ISC frequency was considerably higher and tumor latency shorter for 1833 cells and for 231DD compared to 231. Further, T-ISC frequency was decreased significantly by p27 depletion in 1833 (Fig. 6i–k) and by cJun depletion in 231p27CK-DD

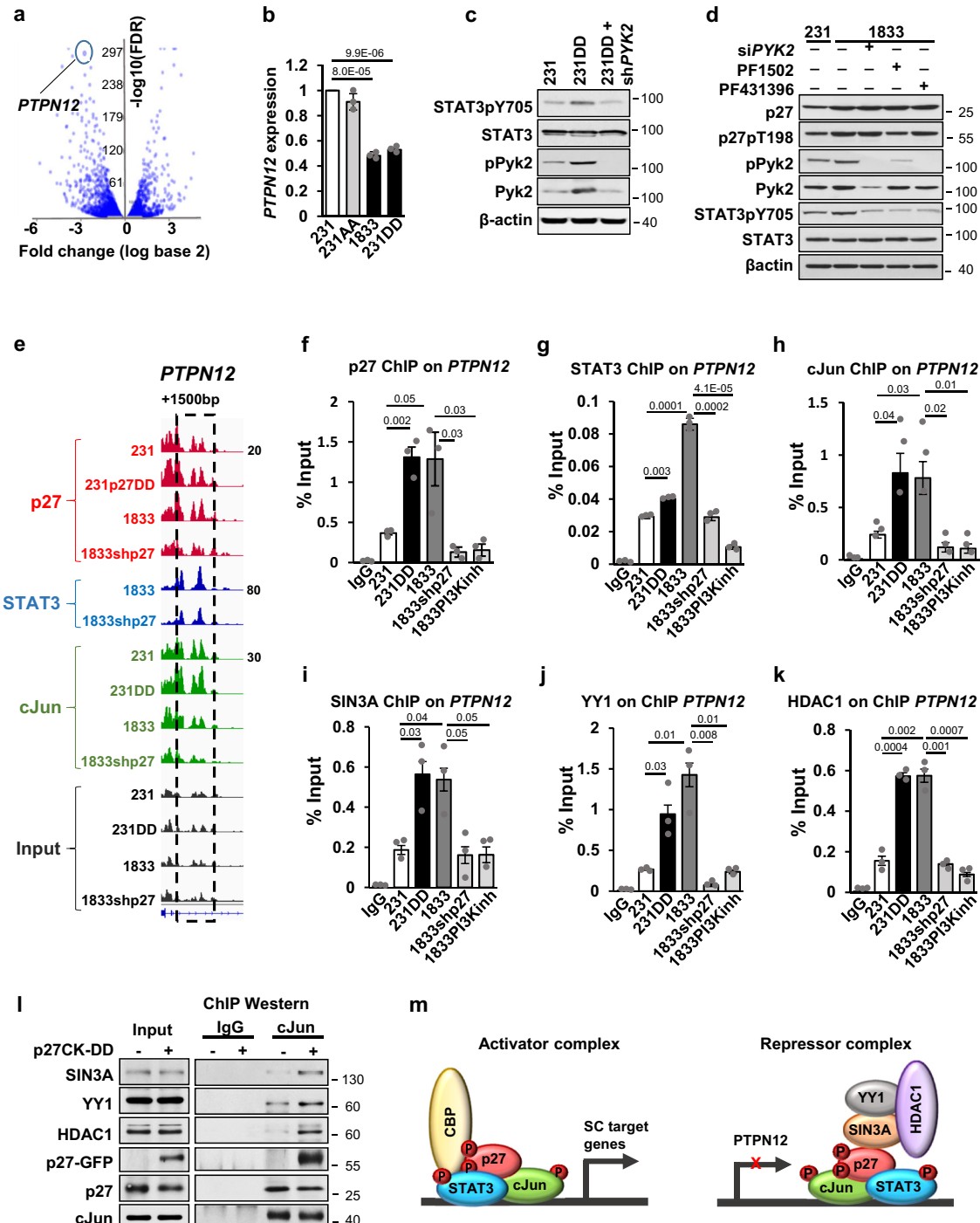

**Fig. 5 | STAT3/cJun/p27 co-repress *PTPN12* in complex with YY1, SIN3A, and HDAC1. a** Volcano plot of genes differentially expressed in 231DD vs. 231. **b** *PTPN12* mRNA expression by qPCR in the indicated lines. **c** Western shows effects of p27CK-DD on total and activated STAT3 and Pyk2 and effects of *PYK2* loss on STAT3 levels and STAT3pY705. **d** Western shows effects of p27pT198 (reduced by PI3K inhibitor, PF1502) and Pyk2 loss (si*PYK2*) and Pyk2 inhibition by PF431396 on STAT3 levels and activation (STAT3pY705) **e** p27, STAT3, and cJun co-occupied peaks at +2 kb of the *PTPN12* TSS. **f**–**k** ChIP-qPCR shows p27 (**f**); STAT3 (**g**); cJun (**h**);

SIN3A (**i**); YY1 (**j**) and HDAC1 (**k**) binding at +2 kb of the *PTPN12* TSS where p27 peaks overlap with binding peaks of STAT3 and cJun. **l** cJun ChIP-Western blot (right) shows interaction on chromatin of the indicated proteins. Input is shown on the left. **m** Schematic representation for p27/STAT3/cJun in activator (left) and repressor (right) complexes. All graphs show mean ± SEM from *N* = 3 biological replicate assays. *p*-values were calculated by paired one-tailed Student's T Test and are indicated in graphs. Source data are provided as a Source data file.

(Fig. S5h, i). Thus, C-terminally phosphorylated p27 increases tumor-initiating cell abundance in a cJun-dependent manner in vivo in this model.

PTPN12 depletion in 231 significantly increased T-ISC abundance (Fig. 6i), while PTPN12 overexpression in 1833 decreased T-ISC (Fig. 6j).

Pyk2 depletion in 231DD abrogated the gain in T-ISC abundance conferred by phosphomimetic p27CK-DD and prolonged tumor latency (Fig. 6i, k). Quantitation of T-ISC frequency is shown in Fig. 6k. Taken together, these data support a model in PI3K-activated cancers, in which C-terminally phosphorylated p27 activates T-ISC expansion or

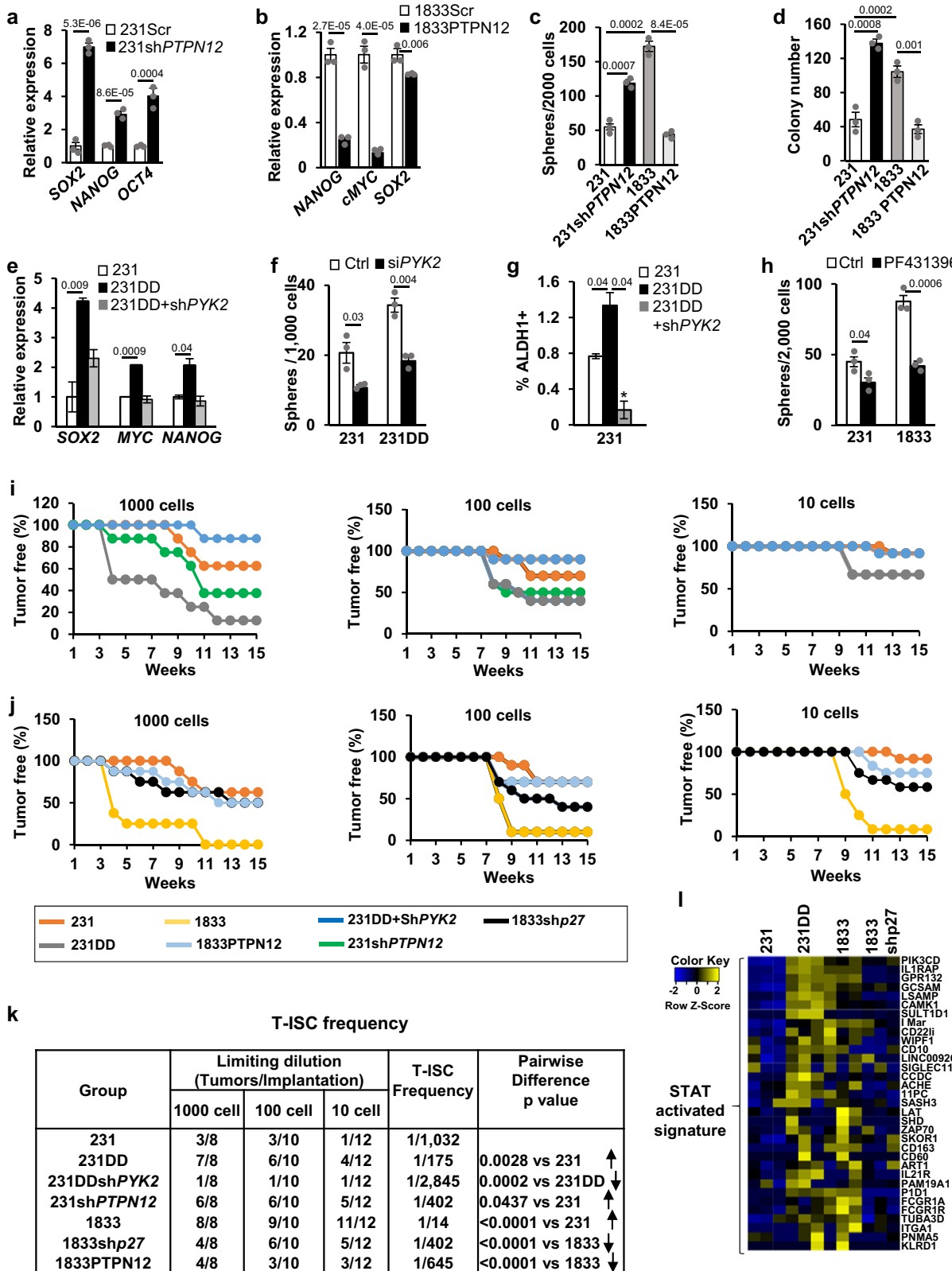

maintenance though p27/STAT3/cJun-dependent PTPN12 loss and activation of Pyk2 and STAT3.

Using the Cancer Genome Atlas (TCGA)/The Cancer Proteome Atlas (TCPA) data, we selected primary human breast cancers in the top quartiles of pSTAT3 and p27pT157 expression on proteomic analysis, and then identified gene profiles common to these. This pSTAT3/p27pT157-activated gene expression profile identified in human breast cancers was strongly overexpressed in 231DD and 1833 compared to 231 and 1833shp27 (Fig. 6l), further validating the relevance of our p27/STAT3 driven gene expression in human disease.

**Fig. 6 | p27 increases stem-like cells in vitro and tumor-initiating stem cell in vivo through loss of *PTPN12* and activation of Pyk2. a, b** Effects of stable *PTPN12* knockdown in 231 (**a**) and *PTPN12* overexpression in 1833 cells (**b**) on embryonic stem cell transcription factors (ES-TFs) expression. **c, d** Effects of *PTPN12* knockdown in 231 and PTPN12 overexpression in 1833 on sphere formation (**c**) and colony formation (**d**). **e, f** Effects of *PYK2* knockdown on p27-driven increased in ES-TFs expression (**e**) and mammosphere formation (**f**). **g** ALDH assay in 231, 231DD, and in 231DD-siPYK2. **h** Effect of Pyk2 kinase inhibitor (PF431396) on sphere formation. All graphs show mean ± SEM from *N* = 3 biological replicate assays. *p*-values were identified by paired one-tailed Student's T Test and shown in graphs. **i, j** Limiting dilution assays of tumor-initiating stem cell abundance in vivo are graphed showing the percent of tumor-free mice over time for the indicated cell conditions. **k** The indicated cell numbers were injected in the mammary fat pad of Balb/c nude mice at limiting dilutions of 231 vector controls, 231DD, 231DD + shPYK2 and 231shPTPN12; and 1833, 1833shp27, and 1833PTPN12. Tumor-initiating stem cell (T-ISC) frequency is shown. T-ISC frequency was calculated by L-Calc Limiting Dilution Software (http://www.stemcell.com/en/Products/All-Products/LCalc-Software.aspx) from STEMCELL Technologies. **l** Gene expression heatmaps show increased expression in 231DD and 1833 of pSTAT3/p27pT157-activated gene profiles derived from primary human breast cancers. Source data are provided as a Source data file.

## p27CK-DD increases mammary duct morphogenesis, hyperplasia, and yields invasive mammary cancers that metastasize

The effects of transgenic expression of p27CK-DD on mammary development and tumor formation were evaluated in a p27CK-DD X MMTVCre bigenic mouse model and compared with that of p27CK- X MMTVCre and MMTVCre only controls (ES targeting vectors shown in Fig. S6a). Carmine red staining of mammary gland (MG) whole mounts at 18 months showed mammary duct hyperplasia and an increased mammary duct branching area in p27CK-DD bigenic compared to p27CK- bigenic, and MMTVCre control mice (Fig. 7a, quantitation graphed in b and duct branching in S6b).

Since the mice did not develop palpable tumors, they were followed for 18 months before histologic evaluation of the 4rth inguinal MG (the largest of the 10 MGs). Mammary TG p27CK-DD and to a lesser extent mammary TG p27CK- both developed microscopic invasive cancers (neoplastic proliferations that extended beyond ductal boundaries into mammary fat) by 18 months. Microinvasive mammary cancers were seen in more of the p27CK-DD mammary TG mice (*n* = 12/14 MG) and more lesions were seen within these glands than in p27CK-bigenics (*n* = 6/14 MG) or in control MMTVCre mice (*n* = 3/14 MG). Rare development of these lesions is normal at 18 months. The % of mice with mammary neoplasia is graphed in Fig. 7c; see also Fig. S6c). In tumor-bearing mice, the mean tumor multiplicity was 4 tumors/MG in MMTVCre controls, 5 tumors/MG in p27CK- bigenics and 11/MG in the p27CK-DD mammary TG mice. Histopathology is shown in Fig. 7d and Fig. S6d–f. The mammary TG p27CK-DD tumors showed strong staining for the transgenic phosphomimetic p27 with anti-p27pT198 and for cytokeratin (representative staining Fig. 7d, for mice 1668 and 56) and the lesions were widespread within the MG. In Mouse 1668, less differentiated areas within the MG tumors showed loss of cytokeratin staining, but persistent staining of the transgenic p27CK-DD. Staining for the myoepithelial marker, p63, confirmed these cancers are invasive. In contrast to the uniform perimeter staining seen in normal and hyperplastic ducts Fig. S6e, the cancers show disrupted basement membranes and invasion into fat Fig. 7d and Fig. S6e. Notably, a 2 mm cancer in the 4th inguinal MG of p27CK- mouse 26 involved and invaded beyond an intramammary lymph node (Fig. 7d). Positive staining for cytokeratin confirmed cancer in the node. p27pT198 was focally positive, consistent with focal phosphorylation/activation of the TG p27CK- product as would be present with local AKT activation in a cancer.

Notably mammary cancers in p27CK-DD mice were not only distributed abundantly throughout affected MG, but also developed liver metastases. 6 of 12 mammary TG p27CK-DD tumor-bearing mice showed micrometastases to the liver (shown histologically for p27CK-DD TG mice 1668 in Fig. 7e and for mouse 61, 76, and 56 in Fig. S6f). Microscopic metastases were detected histologically throughout these livers, with invasive cancer cells detectable both histologically and confirmed by cytokeratin staining, and by staining of the phosphomimetic TG product with p27pT198 antibody at multiple locations throughout the liver tissues (representative data, Fig. 7e). Liver tissue was also positive by qPCR for expression of the transgene in the livers of these mammary TG p27CK-DD mice but not in the livers of Cre control mice (Fig. S6g). The MMTVA promoter drives Cre expression and expression of the floxed p27TG only in MG, salivary gland, and skin, thus TG detection in the liver confirms metastatic spread in all of the p27CK-DD TG MG cancer-bearing mice (qPCR data in Fig. S6g).

## Discussion

Cancer stem cells (CSCs) have emerged as drivers of cancer recurrence[1,10]. CSCs exhibit greater resistance to chemotherapy and contribute significantly to cancer progression, relapse, and drug resistance[1]. Conventional chemotherapy or radiation therapy has proven ineffective in eliminating CSCs. The heterogeneity of CSC populations and their dynamic nature pose further challenges to effective CSC targeting, necessitating a deeper understanding of the molecular mechanisms of CSC maintenance.

In this work, we introduce p27 as an important regulator of CSC expansion. p27 restrains normal growth via cyclin-CDK inhibition. In cancers, p27 is either decreased, or it accumulates in both cytoplasm and nucleus upon C-terminal phosphorylation by PI3K activated kinases, and both associate with poor cancer patient prognosis[54,55]. T198 phosphorylation of p27 facilitates its interaction with RhoA and RhoA-ROCK1 inhibition[24], which increases cancer motility and invasion[56]. p27pT157pT198 also increases Cyclin D-CDK4 assembly and nuclear translocation in early G1[57,58]. However, these gain of function mechanisms are not sufficient to fully explain the pro-oncogenic effects of p27.

Here, we show C-terminally phosphorylated p27 acts to expand or maintain cancer cells with stem cell features in vitro and in vivo. Immortal, non-malignant MCF12A mammary epithelial cells expressing p27CK-DD acquire expression of CSC markers, CD44+ CD24low/-, and the ability to form tumor spheres and colonies in soft agar. p27CK-DD transduction and high endogenous p27pTpT also increase ES-TF expression and sphere formation in both ER+ and ER- breast cancer lines and in the UMUC3 bladder cancer model, supporting the broader relevance of these findings. Finally, p27 phosphorylation also increased tumor-initiating stem cell abundance via *PTPN12* repression, and Pyk2 driven STAT3 activation in vivo. The marked effects of Pyk2 loss on ALDH1 and tumor initiation in these models suggest that its effect on stem cell function might also involve other pathways in addition to p27.

Genetic experiments have suggested that p27 has roles beyond the cell cycle to govern development. p27 null mice exhibit an overgrowth phenotype with hyperplasia of multiple organs and frequent pituitary tumors[59–61]. They exhibit disorganization and expansion of the generative layers of the retina[61] and increased lineage committed hematopoietic precursors[59,62]. In mice and *xenopus*, p27 regulates neuronal differentiation and migration in the cerebral cortex[28,29,32]. In *xenopus*, the p27 homolog, Xic1 also governs myogenesis[26,31] and is required for cardiomyocyte differentiation[27]. p27 also genetically interacts with p107 to control murine chondrocyte maturation[30]. Since p27CK-, that cannot bind to cyclin-CDKs, can compensate for p27 loss to restore neuronal, myogenic, and cardiomyocyte differentiation,

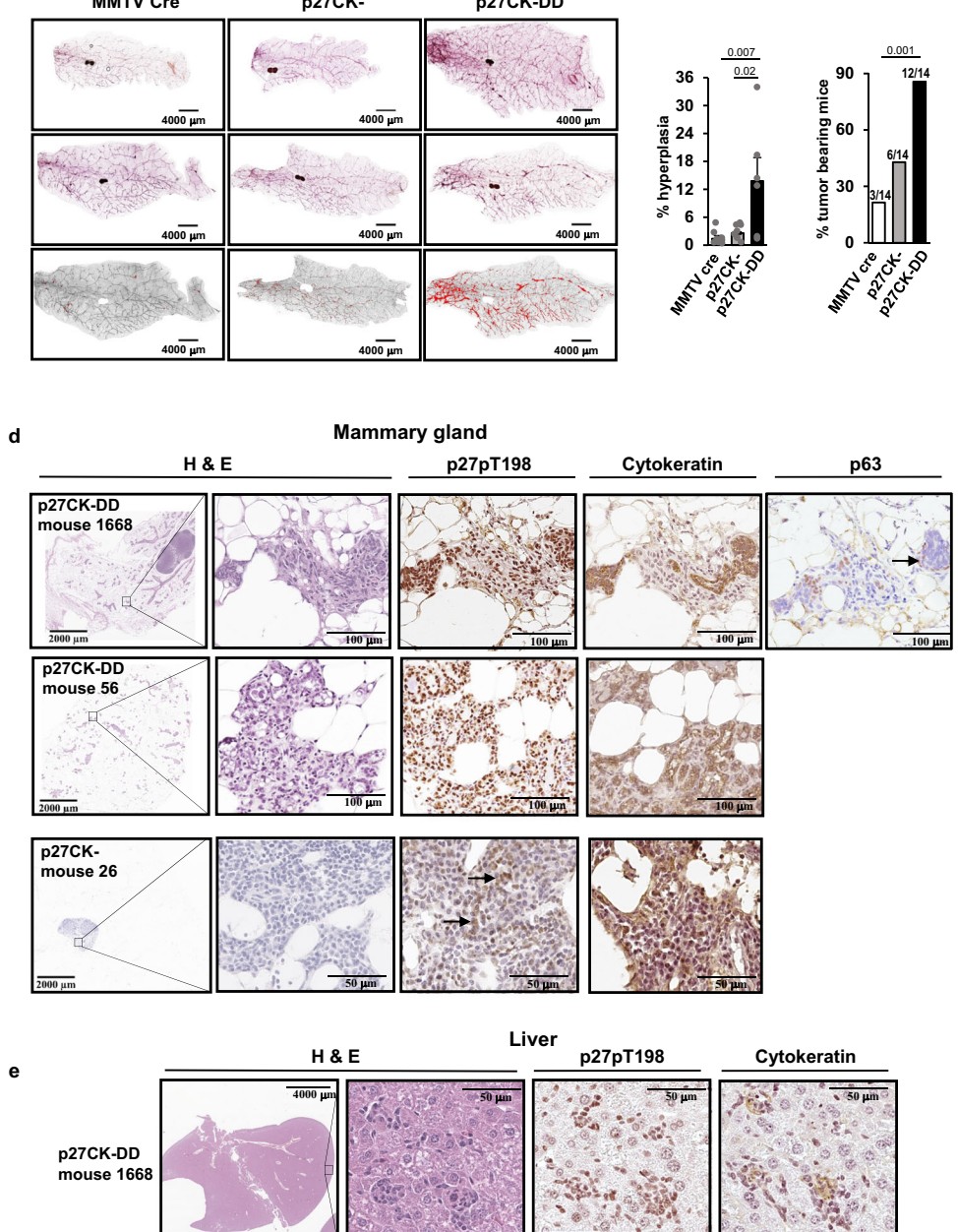

**Fig. 7 | Mammary transgenic expression of p27CK-DD increases mammary duct branching, mammary hyperplasia and yields mammary cancers that generate liver metastases. a** Representative mammary gland whole mounts stained with Carmine red from MMTVCre controls and MMTV-Cre X TGp27CK- (p27CK-) and MMTV-Cre X TGp27CK-DD (p27CK-DD) mice show greater duct branching (top 2 panels) and hyperplasia (bottom panels) in the p27CK-DD bigenics. **b** Duct hyperplasia was quantitated from whole mounts as described and graphed as mean %+/−SEM. p values were calculated by one-way ANOVA with post hoc 2 by 2 comparisons using Tukey correction. $N = 8$ mice for MMTVCre controls, $N = 7$ mice for TGp27CK-, $N = 6$ mice for TGp27CK-DD (See also SFig. 6b for quantitation of duct branching). **c** The number of mice with mammary cancers over the total number evaluated for each genotype is graphed as %. p values calculated by Fisher's Exact Test are shown. $N = 14$ mice for MMTVCre controls, $N = 14$ mice for TGp27CK-, $N = 14$ mice for TGp27CK-DD. **d** FFPE mammary glands from the indicated genotypes were stained with hematoxylin and eosin (H&E) or with antibodies to p27pT198, to cytokeratin or to myoepithelial marker, p63, as indicated and representative photomicrographs shown. **e** Livers show representative micrometastases from mammary cancer-bearing MMTV Cre X p27CK-DD mouse 1668 (see also Fig. S6). Source data are provided as a Source data file.

these noncanonical p27 functions are at least partly independent of its known ability to inhibit cyclin-CDKs.

Knock-in of p27CK- in p27 null mice disrupted the balance of self-renewal and differentiation, increasing lung bronchoalveolar stem cells, and led to lung, pituitary, retinal, and ovarian tumors and lymphoma[23], further supporting the notion that p27 acts through CDK-independent mechanisms. Despite an abundance of genetic data from various model systems that p27 modulates development

in many tissues, the mechanisms underlying this have remained obscure. Present work provides valuable insight into the role of p27 in development. Our data demonstrate that p27 governs mammary gland development. Our mammary transgenic p27CK-DD mice show altered mammary development, with excess duct branching and hyperplasia, leading to invasive and metastatic mammary cancers, indicating a CDK-independent function of C-terminally phosphorylated p27 to expand mammary progenitors. These findings align with

our observation that C-terminally phosphorylated p27 can increase CSCs.

Mechanistically, we show that p27 induces gene drivers of SC and CSC self-renewal by directing STAT3 transcriptional activity. STAT3 is constitutively activated and predictive of poor prognosis in many human malignancies[63] and is required for embryonic, mammary and cancer stem cell self-renewal. Deletion of STAT3 from normal basal mammary cells reduces primary transplantation capacity and attenuates luminal progenitors and duct morphogenesis in mice[64]. In growth arrested MEFs, p27 interacts with E2F4 and p130 to repress target gene expression[53]. C-terminally phosphorylated p27 also co-regulates cJun to activate gene programs associated with EMT and metastasis in cancer cells[33].

Here, we show that p27 drives CSC expansion at least in part through action on STAT3. p27 knockdown reduces global STAT3 chromatin recruitment and enrichment of both STAT3 and p27 at common target genes increased with C-terminal p27 phosphorylation. Most p27/STAT3 co-targets include overlapping cJun binding sites. Notably, these p27/STAT3/cJun co-targets critically govern SC expansion programs and include ES-TFs (*MYC, SOX2, OCT4,* and *KLF4*), the Notch ligand, *JAG1* and the Wnt pathway TF, *LEF1*. cMyc is an embryonic stem cell transcription factor whose dysregulation mediates malignant cell proliferation, immortalization, angiogenesis, metastasis, and CSC self-renewal[65–67]. *MYC* is frequently overexpressed in stem cell enriched basal-like TNBCs[68–70]. The Notch family critically regulates mammary development[71] and CSC maintenance[72,73]. Overexpression of Jag1, a major Notch ligand, confers CSC-like properties, regulating CSC expansion and mammosphere formation in TNBC models[74,75]. Recruitment of cJun, STAT3, the histone acetyl transferase CBP, histone H3K27 acetylation and the expression of these CSC regulatory genes were all decreased by p27 knockdown, revealing a mechanism whereby p27 induces critical drivers of CSC self-renewal.

Resistance to detachment-induced cell death or anoikis is a hallmark of cancer and is required for metastasis. While normal epithelial cells undergo anoikis upon detachment from the basement membrane, stem-like cells are anoikis resistant[76–78], permitting sphere formation and tumor initiation in vivo[79]. Present work identifies the mediator of anoikis resistance, *ANGPTL4*, as a p27/STAT3/cJun co-target gene, essential for p27CK-DD driven ES-TF expression and sphere formation, revealing a role for this gene in CSC maintenance.

These data reveal a dual role for p27 as co-regulator of STAT3/cJun—in which p27 can either up or downregulate target gene expression. While p27-upregulates stem cell pathway genes, p27/STAT3/cJun complexes also downregulate gene drivers of differentiation. Stem cell self-renewal is in many ways the transcriptional converse of differentiation and ES self-renewal is maintained through repression of differentiation genes[80]. Prior work suggested that 27 can serve as a corepressor with E2F4, SIN3A, p130, and HDAC1[52,53]. Not only does p27pTpT downregulate gene programs critical for stem cell differentiation, p27 phosphorylation promotes p27/STAT3/cJun co-recruitment with YY1, SIN3A, and HDAC1 to repress a previously undescribed CSC repressor, the phosphatase, *PTPN12*. *PTPN12*-repressive complexes were increased in cells with high endogenous p27pTpT or phosphomimetic p27, and decreased by p27 knockdown. PTPN12 is a tumor suppressor, whose frequent loss in aggressive breast cancers activates HER2 and EGFR and accelerates breast carcinogenesis and metastasis[48]. Here, we show that PTPN12 also restricts CSC abundance. *PTPN12* repression facilitates STAT3 activation by Pyk2 and ES-TF expression, and the observed p27-driven increases in mammospheres and in tumor formation by breast cancer initiating cells in vivo. Pyk2 not only activates STAT3, a known mediator of ES[80] and cancer stem cell self-renewal[10], but STAT3, once activated would feed forward to induce *PYK2* gene expression[81] and maintain stem cell expansion. The relevance of this pathway to breast cancer is reflected in the coordinate activation of p27-regulated genes within a STAT3-activated gene signature from TCGA cohort of primary human breast cancers.

PI3K/Akt signaling maintains embryonic stem cell pluripotency[82] and malignant SCs in sarcoma and glioblastoma[83]. Here, we identify an additional mechanism whereby PI3K/AKT drives CSC expansion. p27pTpT, generated by PI3K-activated kinases[19,20,22], plays a CDK-independent role to expand cancer stem cells (CSCs). In PI3K-activated cancers, p27pTpT plays a transcriptional regulatory role to co-activate STAT3 and cJun to govern gene programs critical for CSC self-renewal.

## Methods

### Resource availability
This research complies with all relevant ethical regulations approved by Georgetown University and University of Miami.

### Experimental model and subject details
**Cell lines.** The MCF12A cell line was purchased from ATCC (CRL-10782) and cultured as described[34]. MCF7 cell line was obtained from Marc Lippman and cultured in Improved MEM with 10% fetal bovine serum. MDA-MB-231/1833 were from J Massague (MSKCC, New York, NY, USA), 231p27CK-DD, and 1833shp27 grown in Dulbecco's modified Eagle medium (DMEM; Gibco, Grand Island, NY, USA) with 10% fetal bovine serum, 1% glutamine, 1% sodium pyruvate (Invitrogen, Carlsbad, CA, USA)[39,84]. The bladder cancer lines UMUC3 (parental, low metastatic), UMUC3-LuL2 (highly lung metastatic derivative), UMUC3p27CK-DD, and UMUC3-LuL2shp27 were obtained from Dan Theodorescu and grown in Iscove's modified Eagle's medium with 10% fetal bovine serum[85]. All cell lines were grown at 37 °C. Cells were tested for mycoplasma monthly and before in vivo experiments (Sigma-Aldrich, MP0025-1KT). The authenticity of the cell lines was confirmed in 2016, and 2021 through short-tandem repeat (STR) fingerprinting analysis by Georgetown University Tissue Culture Shared Resource (TCSR).

**Animal models.** pCAG-LSL-RFPp27CK- and pCAG-LSL-RFPp27CK-transgenes were microinjected into the C57B6 mice zygote pronuclear in the Animal Models Shared Resource of University of Miami. Founder CAG-LSL-RFPp27CK- transgenic mice and founder CAG-LSL-RFPp27CK-DD transgenic mice were selected and their genotypes were verified by PCR amplification of a 350 bp DNA fragment spanning the RFP and p27 sequences. To activate transgenic expression of p27CK- and p27CK-DD, the pCAG-LSL-p27CK-(KpnI-) and pCAG-LSL-p27CK-DD(KpnI-) transgenic mice were bred with MMTV-CreA mice (purchased from Jackson lab). The genotypes of bigenic MMTV-Cre; p27CK- or MMTVCre; p27CK-DD progeny were verified by PCR analysis of genomic DNA from tail biopsies using Cre-specific and either RPF or p27 primers. Virgin female mice were recovered at 18 months of age. All mice were housed and bred in accordance with institutional guidelines on a 12 h light/dark cycle with constant ambient temperature (22–24 °C) and humidity (46–48%). All murine experiments complied with our IACUC maximum tumor size of 1000 mm³ and the maximal tumor size was not exceeded. Animal protocols were reviewed and approved by the Institutional Animal Care and Use Committee (IACUC) of the University of Miami (protocol 17-166) and of Georgetown University (protocol 2021-0014).

### Method details
**Transgenic vector construction.** Human p27 mutant cDNA encoding p27CK-, with four point mutations in the cyclin/CDK binding domain (R30A, L32A, F62A, F64A), that abolish cyclin/CDK binding ability of p27, was provided by Dr. Steven F. Dowdy in a pEYFP-C1 vector (pEYFP-C1-p27CK-). p27CK- C terminal phosphomimetic mutation T157D and T198D was generated by site directed mutagenesis[34] to create p27CK-T157D/T198D (hereafter p27CK-DD). The RFP fragment (tdTomato gene) was inserted to the p27 vector, creating the plasmid

pRFP-p27CK- and pRFP-p27CK-DD. To create the inducible p27CK- and p27CK-DD transgenes, a CMV enhancer-chicken β-actin promoter sequence and a loxP-neomycin resistance-triplepolyA-loxP "stop" cassette (a kind gift of Drs. Dario Acampora and A Simeone, Institute of Genetics and Biophysics, Andre L.P. Tavares et al. 2015) was inserted upstream of the transgene sequences resulting in pCAG-LSL-RFPp27CK- and pCAG-LSL-RFPp27CK-DD plasmid.

**Vectors and transfection.** Generation and growth of p27 phospho-mimetic mutant, GFP-p27CK-T157D/T198D (p27CK-DD), expressing MDA-MB231, UMUC3, and MCF12A cells was described[34]. In brief, using Lipofectamine Plus, lentivirus vectors carrying three distinct shRNAs targeting CDKN1B (p27) or Plvx-AcGFP, Plvx-AcGFP-p27CK-DD were co-transfected into Lenti-X 293T cells along with Delta VPR and CMV VSVG plasmids (Addgene). After 48 h, viral supernatants were harvested and concentrated through ultracentrifugation at 22,000 RPM and 4 °C for 2 h. to establish stable cell lines expressing either GFP or GFP-p27CK-DD, MCF-12A, MCF-7, MDA-MB-231, and UMUC3 cells were infected with the virus in the presence of 10 μg/ml polybrene. Likewise, stable cell lines expressing either a Scramble control or shp27 were generated using MDA-MB-1833 and UMUC3-LuL2 cells. Lentiviral vectors expressing PTPN12 shRNA (Applied Biological Materials, 38148091) were generated, and transfections yielding 231shPTPN12, UMUC3shPTPN12, and 1833PTPN12 were performed as described[40]. pMD-PTPN12 cDNA clone was obtained from Sino Biological Inc. (HG11556-UT). Confirmation of stable cell lines was achieved by visualizing 100% GFP expression and performing Western blot.

**siRNA-mediated knockdown.** Oligonucleotide siRNAs to *STAT3*, *ANGPTL4*, *PYK2*, *PTPN12*, and scrambled controls were purchased from ThermoFisher (4392420, 4390824, 4390843, 4390849) and transfected according to manufacturer using Lipofectamine RNAiMAX (Fisher Scientific, 13778075). The oligonucleotides sequences are not provided by the manufacturer.

**CRISPR-mediated STAT3 knockout.** Transfections of 1833 cells by HDR-mediated STAT3 human gene knockout kit (OriGene, KN204922RB) were done with TurboFectin 8.0 transfection reagent (OriGene, 13778075) per manufacturer's instructions. STAT3 lost was confirmed by qPCR and WB.

**Treatment of cells.** For qPCR, western blots, and sphere assay with inhibitor treatments, cells were treated with 250 nM PI3k inhibitor (Pfizer), 500 nM Pyk2 inhibitor (Sigma-Aldrich, PZ0185), and 30 μM STAT3 inhibitor (Sigma, D4071-10MG). Inhibitor treatment was done for 48 h. For STAT3 ChIP-seq, cells were treated with 10 nM human recombinant IL6 (R&D Systems, 206-IL-010) dissolved in PBS + 0.1% bovine serum albumin (BSA) for 30 min.

**Western blotting and Immunoprecipitation Westerns.** Cells were washed with PBS, and lysed in RIPA buffer (ThermoFischer, 89900) resolved by SDS–PAGE, and transferred to a polyvinylidene difluoride membrane. The membranes were incubated with the indicated primary antibodies and HRP-conjugated secondary antibodies. The immune-reactive bands were visualized using a chemiluminescent substrate (ThermoFisher; 32106), and were exposed to X-ray film (Denville, Holliston, MA, USA). Primary antibodies for p27 (610241, 1:5000 3686 s, clone D69C12, 1:1000) from Transduction Labs and from Cell Signaling; for p27pT198 (AF3994, 1:2000) from R&D Systems; for STAT3 (9139 s, clone 124H6, 1:1000), STAT3Y705 (9145 s, clone D3Z2G, 1:2000), PYK2 (3480 s, clone 5E2, 1:1000), pPYK2 (3291, 1:1000), CBP (7389, clone D6C5, 1:1000), p130Cas (13846, clone E1L9H, 1:1000), p-p130CAS (4011, 1:1000), Myc (5605 s, clone D84C12, 1:1000), Sox2 (3579 s, clone D6D9, 1:1000), Nanog (8822, clone D2A3,

1:1000), cJun (9165 s, clone 60A8, 1:1000) and cJunpS63 (2361 s, clone 54B3, 1:1000) were obtained from Cell Signaling; for β-actin (A1978, clone AC-15, 1:5000) from Sigma-Aldrich; and for PTPN12 (14735, 1:1000) from Abcam. Secondary antibodies: Anti-rabbit IgG HRP-conjugated (Promega, W4011, 1:10,000) and Anti Mouse IgG HRP-conjugated (Promega, W4021, 1:10,000). For Immunoprecipitation Westerns, cells were lysed with Cell Lytic M (Sigma, C2978). 1 mg cell lysate was incubated overnight at 4 °C with 2 micrograms p27 antibody (Cell signaling, 3686s, clone D69C12) or control IgG (Cell signling, 3900s). Complexes were collected on Pierce protein A/G plus agarose (Thermo Scientific, 20423) and then resolved by SDSPAGE, transferred to PVDF membrane and immunoblotted.

**Quantitative real-time PCR (q-PCR).** Total RNA was extracted from cell lines with RNeasy Plus Mini kit (Qiagen, 74134), according to the manufacturer's instructions. cDNA was synthesized from total RNA (1 μg) using iscript cDNA synthesis kit (Biorad, 1708891). qPCR was performed using SYBR green (Biorad, 1708882) on Light Cycler 480 (Roche) and quant studio 6 pro (Applied Biosystems by Thermo Fisher). The fold changes were normalized using GAPDH, and each reaction was conducted in triplicates. All qPCR reactions were repeated in at least 3 independent biological assays. Primer sequences for each gene are listed in Table S1. For ES-TFs, all of *SOX2, NANOG, MYC, KLF4*, and *OCT4* were assayed, but only positive data were graphed.

**RNA sequencing (RNA-seq) assay.** The quality of total RNA isolated from the cell lines was measured using Bioanalyzer RNA Nano 6000 (Agilent Technologies, Santa Clara, CA, USA). Library preparation was performed by TruSeq Stranded Total RNA Library Prep (Illumina, 1000000040499), and the quality was confirmed using KAPA qPCR Library Quantification (Kapa Biosystems). Paired-end sequencing was performed on Illumina 150 cycles 400 M reads. Quality of raw sequence reads was assessed using FastQC. Trimmomatic[86] was used to trim sequences of bad quality bases, adapter, and primer sequences. Genome alignment was performed using STAR aligner[87], with UCSC human genome (hg38). After the read alignment, raw counts were estimated using featureCounts[88]. DESeq2[89] was applied to identify differentially expressed genes between groups of samples. Pathway and gene set analysis was performed using Gene Set Enrichment Analysis (GSEA)[90]. RNA-seq was performed on three independent biological replicates[33].

**Chromatin immunoprecipitation (ChIP) assay.** ChIP assay was performed by ChIP-IT High Sensitivity® (HS) Kit (Active motif; 53040) according to kit's instruction. Briefly, after cells were crosslinked with 1% formaldehyde, the cells were lysed and sonicated. For the immunoprecipitation, sonicated samples were incubated with p27 (Cell Signaling; 3686 s, clone D69C12, 2 μg), STAT3 (Cell Signaling; 12640 s, clone D3Z2G, 2 μg), cJun (Cell Signaling; 9165s, clone 60A8, 2 μg), CBP (Cell Signaling; 7389 s, clone D6C5, 2 μg), H3K27ac (Cell Signaling; 8173 s, clone D5E4, 2 μg), SIN3A (Abcam; ab3479, 2 μg), YY1 (Abcam; ab12132, 2 μg), HDAC1 (Santa Cruz; sc-7872, clone H-51, 2 μg), IgG Rabbit (Cell Signaling; 2729 s, 2 μg), and IgG mouse (Santa Cruz; sc-2025, 2 μg) antibodies overnight at 4 °C. Samples were collected on A/G plus-agarose. DNA was eluted after washing of the beads and decross-linking. Eluted DNA was used for qPCR. STAT3 ChIP DNA was also used for sequencing. The ChIP DNA library was made using NEBNext® Ultra™ II DNA Library Prep with Sample Purification Beads kit (NEB; E7103S) per kit's instruction. ChIPseq was performed on a single biological replicate and the ChIPqPCR validations were conducted on three independent biological replicates in which each biological replicate contained three technical repeats. Primer sequences are listed in Table S1. p27 and cJun ChIPseq data in our cell models was from Yoon et al.[33].

**ChIP-seq analysis.** ChIP-seq analysis for STAT3 in 1833 and 1833shp27 was compared with ChIP seq for p27, and cJun in 231, 231DD, 1833, and 1833shp27 published in Yoon et al.[33], and was setup following ENCODE3 ChIPseq pipeline (https://www.encodeproject.org/chip-seq/transcription_factor/). Initial quality was inspected by FastQC (version 0.11.8) (https://www.bioinformatics.babraham.ac.uk/projects/fastqc/), and adapters were trimmed with Trim Galore (version 0.4.4) (https://www.bioinformatics.babraham.ac.uk/projects/trim_galore/) leaving preprocessed reads for global alignment. Genome alignment was performed by bowtie (version 2.3.3.1) (https://bowtie-bio.sourceforge.net/bowtie2/index.shtml) against the hg19 version of the human genome. After alignment, unaligned reads were removed from the alignment file, and reads that were multimapped were corrected to keep the best alignment. Duplicate reads were marked and removed from the aligned bam file using PicardTools (version 2.1.1) (https://broadinstitute.github.io/picard/). To start assessing quality of alignment, the PCR bottleneck coefficient was calculated based off this final aligned bam file. Further QC was calculated including the cross correlation coefficient with phantom peakqualtools (version 1.2.2) (https://github.com/kundajelab/phantompeakqualtools)[91]. After QC, peaks were initially called with MACS2 (version 2.2.1) (https://github.com/macs3-project/MACS/releases)[92] with general default parameters.

Peaks that belonged to the blacklist region were then excluded from further analysis, and peaks were filtered with q-value threshold of e−10. Overlapping peaks were then calculated using customized scripts based off bedtools (version 2.26.0) (https://bedtools.readthedocs.io/en/latest/)[93] to calculate the reads that belonged to individual samples, and overlapping peaks. Resulting peaks were annotated to the hg19 Ensembl reference using the R package ChipPeakAnno (version 3.17) (https://bioconductor.org/packages/release/bioc/html/ChIPpeakAnno.html)[94,95]. Peaks that were +/−5 kb away from gene transcription start site (TSS) were assigned the gene. Intensity plots were calculated using the deepTools python package (version 2.5.3) (https://deeptools.readthedocs.io/en/develop/index.html)[96] to calculate the coverage matrix and plot the resulting signal intensities. Gene start sites for these plots were set to +/−2 kb of the start site of the peak for the coverage matrix. Peaks were then split out into different categories with 0.75 thresholds of intensity values from the comparison sample. Those with >0.75 were said to be gained, those <0.25 were said to be lost peaks in the comparison with the in between intensities were classified as retained peaks.

**Sphere formation assay.** The cells ($2 \times 10^3$ cells/well) were plated in ultra-low attachment 6-well plates (Corning Inc., 29443-03) with 2 mL of DMEM-F12 media supplemented with 1 µg/ml Hydrocortisone (Stemcell Technologies; 7926), 4 µg/ml Heparin (Stemcell Technologies; 7980), 10 µg/ml Insulin (ThermoFischer; 12585014), 1X B27 (ThermoFischer; 17504044), 20 ng/ml EGF (ThermoFischer; PHG0311), and 20 ng/ml FGF (ThermoFischer; 13256-29) and incubated in culture conditions for 12–21 days. Spheres >75 µm in size were visualized and counted using a GelCount™ Tumor Colony Counter (Oxford Optronix, Abingdon, UK).

**Colony formation assay.** The cells (5000 cells/well) were seeded into 6-well plate with 0.5% and 0.35% of agarose for bottom and upper layers. After incubation for 4 weeks, colonies stained with Thiazolyl Blue tetrazolium bromide (Alfa Aesar, L11939-03) visualized and counted using a GelCount™ Tumor Colony Counter.

**ALDH assay.** The ALDH assay has been done by using ALDEFLUOR™ Kit (Stemcell Technologies; 01700) according to the manufacturer's instruction.

**Flow cytometry.** Cells were incubated with anti-CD44-APC (BD Biosciences, 559942, clone G44-26, 1:5) and anti-CD24-PE (BD Biosciences,

560991, 1:5) antibodies for 60 min at 4 °C. After washing twice with PBS containing 0.1% bovine serum albumin (BSA), the cells were analyzed using a BD FACSCanto II Flow Cytometer (BD Biosciences).

**STAT3/p27pT157 activated gene expression profile.** All patient data are derived from The Cancer Genome Atlas breast cancer (TCGA-BRCA) data set, comprising 179 ER negative breast cancers. Clinical and RNA-seq data for these breast cancers were downloaded using R "TCGA-biolinks" package[97]. Reverse phase protein lysate array phosphoprotein data (normalized data: Level 3) for these patient tumors were downloaded from the data portal on TCPA website (http://tcpaportal.org/tcpa/). Breast cancers within the top quartiles of both STAT3_pY705 (pSTAT3-high) and p27pT157 (p27-high) were grouped together versus all other patients. Genes differentially expressed in tumors with p27-high and pSTAT3 high compared to all others were identified with R DESeq2 package[89]. The expression of 851 differentially expressed genes was compared to our RNA-seq data from 231, 231p27CK-DD, 188 and 1833shp27 lines to identify STAT- activated genes whose expression was elevated in p27 activated lines compared to 231 and 1833shp27.

**Coordinate expression of NOS signature and p27-regulated genes.** The profiles of NOS activated and repressed genes in ESC[45] were compared to our RNA-seq data in all 4 lines. A subset of NOS activated target genes preferentially upregulated in p27 activated lines and a subset of NOS-repressed genes that were also preferentially repressed in p27-activated lines versus 231 and 1833shp27 were displayed in heatmaps and analyzed by GO.

**In vivo limiting dilution T-ISC assay.** All animal research was conducted in accordance with the University of Miami Animal Care Committee. For limiting dilution T-ISC assays, 5-week-old female Balb/c nude mice were purchased from Charles River Laboratories (Boston, MA, USA). Limiting dilutions of 10, 100, and 1000 cells were each suspended in 10 mg/ml Matrigel with Hanks' Balanced Salt Solution (HBSS; Lonza, 10-547F), and injected into the 4th inguinal mammary fat pad ($n = 12$, 10, 8 mice per group). Mice were euthanized per IACUC guidelines. Tumor size was measured twice/week, and tumor volumes were estimated as length × width × width × 0.5. T-ISC frequency was calculated by L-Calc Limiting Dilution Software (STEMCELL™). All murine experiments complied with our IACUC maximum tumor size of 1000 mm³ and the maximal tumor size was not exceeded.

**Whole mounts (Carmine Red) staining.** The fourth mammary glands of mice were spread onto an electrostatically adherent microscopy slide and allowed to air dry for 5 min Glands were then fixed in glacial acetic acid: ethanol (1:3), hydrated, and stained overnight in 0.2% carmine red (Sigma-Aldrich; C-1022). The glands were dehydrated by sequentially submersion in 70%, 80%, 95%, and 100% ethanol for 5 min each, cleared in toluene and mounted. Slides were scanned using Aperio GT450 (automated, high-capacity digital slide scanner). The total duct branching area was calculated using QU path (v0.4.3; https://qupath.github.io) and mammary glands containing hyperplastic areas were analyzed using Image J[98]. (https://imagej.net/ij/download.html) as described[98].

**Hematoxylin and eosin (H&E).** Mammary glands or liver tissues were fixed in 10% neutral buffered formalin for 24 h and then paraffin embedded. Tissue sections were cut at 4 microns. The slides were stained with hematoxylin (VWR; 95057-844), followed by eosin-Y alcoholic stain (Epredia; 6766007) and then mounted and slides were scanned using Aperio GT450 (automated, high-capacity digital slide scanner).

**Immunohistochemistry (IHC).** Immunohistochemistry for phospho-p27 Kip1(Thr198) rabbit polyclonal antibody (ThermoFisher; PA5-

36862, 1:100), pan cytokeratin (Dako; M3515, Clone AE1/AE3, 1:100), and p63 (Biolegend, Clone Poly6190, 1:200) was performed on FFPE mammary glands and indicated livers. For antigen retrieval, the slides were immersed in 10 mM Na-citrate buffer pH 6.5 using IHC-Tek Epitope Retrieval Steamer Set (Fisher Scientific; NC0070392) for 40 min. The slides were treated for 20 min with 0.3% hydrogen peroxide to block endogenous peroxidase activity, washed with PBS then incubated with primary antibody 1 h at room temperature and then incubated with secondary biotinylated goat anti-Rabbit IgG Antibody (H + L) (Vector Laboratories; BA-1000-1.5, 1:200) for 1 h, followed by VECTASTAIN ABC Reagent containing streptavidin conjugated with horse radish peroxidase (HRP) (Vector laboratories; PK-4000) for 30 min. The signal was detected by using 3.3′-Diaminobenzidine (DAB) (Vector laboratories; SK-4100) and then slides were counterstained with hematoxylin. Slides were imaged by scanning using Aperio GT450 (automated, high-capacity digital slide scanner).

**Statistics and reproducibility.** Data are presented as the mean ± SEM. Statistical comparisons between two groups were performed by the paired one-tailed Student's $t$ test. One-way ANOVA analysis followed by Tukey post hoc tests was used to compare more than two groups. A value of $P < 0.05$ was considered significant. Biological replicates refer to experimental replicates and replicates form biologically distinct sources.

For mice experiments, the sample sizes were based on our previous experiences with similar analyses. We used 8–14 mice per experimental group for all studies. A total of 240 mice were used for the TISC experiment, and an additional 42 mice were used for the transgenic studies. No statistical method was used to predetermine sample size. No data were excluded from the analyses. The experiments were not randomized. The Investigators were not blinded to allocation during experiments and outcome assessment.

**Kaplan–Meier analysis.** Kaplan–Meier plots were generated using Kaplan–Meier plotter (https://kmplot.com/analysis/). The KM Plotter software identifies the cut offs for low and high gene automatically as the expression levels that separate good and bad outcome groups with the greatest statistical significance.

### Reporting summary
Further information on research design is available in the Nature Portfolio Reporting Summary linked to this article.

## Data availability
The RNA-seq and ChIP-seq data reported in this paper have been deposited at GEO and are publicly available. All remaining data is available in the Article, Supplementary and Source data files. STAT3 ChIP-seq data generated in this study are deposited in GEO under GSE233351 accession number and are publicly available. To review GEO accession GSE233351: Go to https://www.ncbi.nlm.nih.gov/geo/query/acc.cgi?acc=GSE233351. The RNAseq and the p27 and cJun ChIp-seq data are deposited in GEO under GSE112446 accession number[33] and are publicly available. To review GEO accession GSE112446: Go to https://www.ncbi.nlm.nih.gov/geo/query/acc.cgi?acc=GSE112446. Source data are provided with this paper.

## Materials availability
This study did not generate new unique reagents. Our TGp27CK-DD and TGp27CK- mice are available for distribution by J.M.S.

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

## Acknowledgements

We acknowledge the Animal Models, Flow Cytometry & Cell Signaling, Histopathology and Tissue, Tissue Culture and Biobanking Shared Resources of Lombardi Comprehensive Cancer Center, Georgetown University and the Flow Cytometry, Cancer Modeling, and Onco-Genomics Shared Resources of Sylvester Comprehensive Cancer Center, U of Miami. Work was supported by R01CA253111 (J.M.S., K.B., and L.M.), DOD BCRP W81XWH-17-1-0456 (J.M.S. and K.B.), and in part by R01GM146409 and BCRF SPEC–23-023 (L.M.) and NIH R01GM113256 (K.B.).

## Author contributions

Conceptualization: J.M.S., L.M., K.B., and S.R. Methodology, validation, and data analysis: S.R., H.Y., K.J., M.K., H.M.N., A.B., W.H., M.S., D.Z., and Z.Z. Genomic data analysis: S.R., H.Y., M.K., and D.B. Pathology review of mammary tumors: T.A.I. Quantification of mammary tumors and hyperplasia: M.J. Data curation, funding acquisition, project administration, resources, supervision: J.M.S. Writing—original draft: S.R., H.Y., and J.M.S. Writing—review & editing: All authors.

## Competing interests

The authors declare no competing interests.
