## [Peer Review File · Nature Communications]

Reviewers' Comments:

Reviewer #1:

Remarks to the Author:

In this manuscript, the authors address the role of p27, specifically its C-terminal phosphorylation on Thr 157 and 198, as a mediator of malignant cell behavior. Focusing largely on sublines of the triple negative breast cancer cell line MDA-MB-231 in which phosphorylation of these residues is modulated, they provide evidence that p27 can affect gene expression and cellular phenotypes to promote neoplastic phenotypes. They provide complementary information from mouse models and human databases to further support the proposed mechanisms.

Overall, the data provided are strongly supportive of the mechanisms proposed, particularly with respect to p27 serving as a transcriptional cofactor for STAT3 and cJUN. The use of information from various experimental approaches is a notable strength. The following points should also be considered:

1. On page 12 and in the legend of figure 3, it is stated that p27 is recruiting STAT3, cJUN, and CBP to specific promoters. This would suggest that p27 is binding to these sites first, and then bringing in these other proteins. While p27 may enhance binding of these proteins to these sites, since it presumably lacks DNA binding ability (which STAT3 and cJUN clearly have), the text should probably be clarified in this respect.
2. The authors use the word "unprecedented" two times in the manuscript. In neither case is the use appropriate (that is, uncovering a relationship that has never been shown before).

Reviewer #2:

Remarks to the Author:

In this manuscript, the authors study the kinase-independent role of p27 c-terminal phosphorylation in regulating breast cancer stem cells and cancer progression. While kinase independent roles of p27 have been reported in cancer progression, studying the involvement in cancer stem cells has novelty. They found strong evidence that p27pT157pT198 could regulate mammary stem cells and cancer progression using both cell culture and in vivo limiting dilution transplantation assays. They also discovered a novel mechanistic underpinning for such a role of p27 involving STAT3 and a few other transcriptional co-regulators. However, there are a number of concerns especially in lacking knockin confirmation.

Main concerns:

1. The major conclusion that cancer stem cell (CSC) abundance is transcriptionally regulated by C-terminally phosphorylated p27 (p27pT157pT198) is based on experiments using overexpression of a p27 mutant that is kinase dead and has both T157 and T198 mutated to a phosphomimetic residue (D). But this overexpression strategy is not adequate to prove the main conclusion. The authors should confirm their major findings using crispr knock-in of this phosphomimetic. Furthermore, phosphomimetic gain of function mutants should be accompanied by loss of function mutants (such as alanine mutants that prevents phosphorylation) which should also be knocked into the genome.
2. The authors soften many of their conclusions at the end of experiments or even a series of experiments by using words such as "appear," leading readers to question how confident the authors are in their experimental design and execution. In particular, this reviewer questions the validity of their section title "p27 phosphorylation increases ES-TF expression and sphere forming cell abundance" when the final conclusion statement of this section is "Taken together, these data indicate that p27pTpT or p27CK-DD expression appears to increase stem-like cells and stem cell-related gene expression."
3. The authors state that the "STAT3 binding peaks were significantly reduced upon p27 depletion

(Figure 2A right)", but there is no statistical analysis attached to this statistical term and the cited van diagram shows a very modest reduction (from 38k to 33k).

4. The authors state that "DNA binding motif analysis showed that both STAT3 and Jun/AP1 motifs are frequently found within p27, STAT3 individually bound sites and at p27/STAT3 co-occupied sites (Figure 2H)," but the cited Fig 2H does not appear to show all of the data referenced. Also this panel's lettering is a bit confusing.

5. The cutoff for high expression in KMP in Fig 3 (1212 for high expression and 2439 for low expression) lacks justification. The same is true for other KMP of human data.

6. While it is possible that p27 binds to target genes first and then recruits STAT3 and c-Jun, the data are also consistent with that STAT3 binds first and then recruits p27. So caution should be exercised in interpretation.

7. The use of PI3K inhibitor (PF1502) to link c-terminal phosphorylation of P27 to activation of Pyk2 and Stat3 activation needs to be very cautious. PI3K likely can regulate other substrates that can also affect p27 activity.

8. Justifications should be given to why different subsets of stem cell factors (NANOG, SOX2, OCT4 vs. NANOG, and SOX2, and cMYC,) are assayed for knockdown vs. overexpression and between different cell lines. It is unclear why not all four factors are measured for all treatments to be consistent.

9. The conclusion that "Cancers were more prevalent, and more abundant within the MG from p27CK-DD mammary TG mice (n=9/11 with mammary cancers) (Figure 7C,D)" is problematic. The cited panel C shows small precancerous lesions that are not yet tumors. Only palpable masses should be considered to be tumors and Kaplan-Meier tumor free plots should be presented in Fig 7 to better summarize the tumor data.

10. The Mouse 26 panel in Fig 7 appears to a lymph node rather than mammary lesions.

11. Higher resolution photos in Fig 7 are needed to substantiate the claim for "strong staining for the transgenic p27CK-pT198 protein in both cytoplasm and nucleus"

12. Please ensure that liver mets in Fig 7 are real. The current images appear to show inflammatory cells in liver.

13. The authors stated that qPCR detected transgene expression in liver without showing the data. Please show the data as part of supplementary figs and ensure proper controls to ensure that the signal is due to metastasis but not leaking expression of the transgene. A good negative control is the liver before primary tumor detection.

Minor concerns:

1. Please state in Results whether the p27 and STAT3 ChIP data were generated by the authors for this study or previously published.

2. The statement "p27 might transcriptionally co-activate STAT3 to drive genes governing CSCs expansion" is confusing and may be interpreted to suggest that p27 is involved in transcription of STAT3.

3. Fig 4C is cited for "transduction of p27CK-DD into 231 increases activated phospho-isoforms of both Pyk2 and STAT3 (Figure 4C)." But the actual data appears to be in Fig 5C.

4. There is a punctuation error in the following sentence: "cMyc, is an embryonic stem cell transcription factor whose dysregulation mediates malignant cell proliferation, immortalization,

angiogenesis, metastasis and CSC self-renewal.”

Reviewer #3:

Remarks to the Author:

Razavipour and coworkers describe in this interesting manuscript a novel CDK-inhibitory independent function of the CDK inhibitor p27Kip1 in transcriptionally regulating genes involved in stem cell expansion and self-renewal, leading to cancer stem cell expansion. p27 becomes recruited in complex with STAT3/CBP to genes including MYC, ANGPT4 or JAG1. It also recruits HDAC1/SIN3A complexes to the PTPN12 gene, which leads to activation of Pyk2. To demonstrate physiological significance, the authors use a transgenic mouse model of mammary expression of a CDK/cyclin-binding defective C-terminal phosphomimetic p27, that causes increased mammary duct branching, hyperplasia and invasive metastatic cancer.

These data expand previous reports from the Slingerland lab, which recently used this model to show that C-terminally phosphorylated p27 is a coactivator of cJun and drives tumour progression and metastasis. They also demonstrated that cytoplasmic p27CK-DD binds STAT3, induces Twist1 and promotes EMT.

The present manuscript adds important novel insights in the CDK-independent transcriptional functions of C-terminal phosphorylated p27. One of the main conclusions, that PI3k-signalling alters the transcriptional function of p27 by C-terminal phosphorylation should be tested more directly, by comparing the phosphomimetic mutant p27CK-DD to a nonphosphorylatable mutant p27CK-AA. The mutant CK-AA should also be included in IPs to exclude effects of CDK/cyclin binding/recruitment deficiency of CK-DD on DNA. The cell lines 231 and 1833 might not be ideal for this since the ratio (pT198 p27 vs. total p27) is rather similar in 231 and 1833 cells (Fig. 5D). Prior to publication in Nature Communications, the missing controls discussed below should be included for key experiments.

Main points:

1. Fig. 1A,B: This figure is very similar to a figure shown in the previous paper (Yoon et al., 2019). The authors might comment on the novel information (more regulated genes have been identified) and why some of the previously top scoring pathways (e.g. ECM receptor interaction; Protein digestion and absorption) were not identified in the novel pathway analysis.
2. Fig. 1C: Does expression of EGFP-p27CK-AA to level as EGFP-p27CK-DD lead to no increase of SOX2, NANOG or cMYC? Is there no increase in sphere formation?
3. Fig. 1E,F: T198 phosphorylation increases its stability. Are protein level of p27CK- and p27CK-DD similar? Is p27CK- not phosphorylated on T157 and T198? What is the CD44 expression of CK-?
4. Fig. 1G: There is a scale bar, but its size is missing. How many cells are represented in the “dots” in vector or CK-? Could the colony number of CK- be included in the bar graph?
5. Fig. 2C: Does the p27 DNA-binding heat map change upon PI3k inhibition? Or is DNA binding of p27 not altered upon C-terminal phosphorylation?
6. Fig. 3F: The occupancy of the MYC promotor by p27 does not significantly change between 231 and 231DD. Does this mean that binding is independent of C-terminal phosphorylation? Or is only phosphorylated p27 bound? Also here, PI3K inhibition might change binding. Why is binding of STAT3 generally not shown for 231 and 231DD?
7. Fig. 3H: Does the increase in 231DD reflect the degree of p27 overexpression? A control 231-AA is missing here. This control would also be important for Fig. 3J-L and other p27 ChIP (e.g. Fig. 5F). Without this control, the conclusion on page 9 “Thus p27pTpT appears to induce genes ... by enhancing stable STAT3 and cJun recruitment” might be an overinterpretation.

8. Fig. 3M, Fig. 4C: What antibody was used for the p27 IP? Could it be that the antibody does not precipitate CDK-bound wt p27? It appears that EGFP-p27 is selectively precipitated.

A control of EGFP-p27AA is missing. Without this control, it cannot be judged if p27-DD enhances complex formation. It would be a concern if endogenous wt p27 does not bind endogenous cJun, CBP or STAT3. According to the model of the authors, p27 of 1833 cells should precipitate comparable level of cJun, CBP or STAT3. Has that been tested?

9. Fig. 5B: 231-AA would be an important control.

10. Fig. 5G: In this STAT3 ChIP, the signal in 231DD seems comparable to 231 and 1833shp27 – and much lower than in 1833. Any explanation why?

11. Fig. 5L: The control 231-AA is missing here. In addition to the phosphomimetic mutations, EGFP-p27CK-DD could also act different because it fails to bind CDK/cyclins or the EGFP portion might alter the complex composition.

12. Fig. 6G. shPyk2 reduced ALDH1 activity in 231DD far below levels of 231. Does this mean that shPyk2 might act independent on ALDH1 activity?

13. Fig. 7: T298 phosphorylated p27 has been reported to be much more stable than p27. What are the protein expression level of CK- and CK-DD?

Minor points:

Suppl. Fig. 1G: the labelling of the STAT3 western blot must be swapped.

The figure legend for Suppl. Fig. 1 should be corrected ((G-H shown no colony forming assay...)).

Legend for Fig. 6K (table T-ISC frequency) is missing. The legend for Fig. 6K is for Fig. 6L (heatmap).

Legend for Figure S5 H,I,J: J is included in H; H and I are swapped.

Legend for Figure S6 is missing.

GEORGETOWN UNIVERSITY MEDICAL CENTER

LOMBARDI COMPREHENSIVE CANCER CENTER
Research • Education • Treatment

January 15, 2024

Responses to Reviewers

Please see responses to reviewers below.

The reviewer comments are in regular font and our responses are italicized.

Reviewer #1 In this manuscript, the authors address the role of p27, specifically its C-terminal phosphorylation on Thr 157 and 198, as a mediator of malignant cell behavior. Focusing largely on sublines of the triple negative breast cancer cell line MDA-MB-231 in which phosphorylation of these residues is modulated, they provide evidence that p27 can affect gene expression and cellular phenotypes to promote neoplastic phenotypes. They provide complementary information from mouse models and human databases to further support the proposed mechanisms.

Overall, the data provided are strongly supportive of the mechanisms proposed, particularly with respect to p27 serving as a transcriptional cofactor for STAT3 and cJUN. The use of information from various experimental approaches is a notable strength. The following points should also be considered:

1. On page 12 and in the legend of figure 3, it is stated that p27 is recruiting STAT3, cJUN, and CBP to specific promoters. This would suggest that p27 is binding to these sites first, and then bringing in these other proteins. While p27 may enhance binding of these proteins to these sites, since it presumably lacks DNA binding ability (which STAT3 and cJUN clearly have), the text should probably be clarified in this respect.

Response: On page 12 and in the legend of figure 3, we no longer state p27 is recruiting STAT3, cJUN, and CBP to specific promoters, rather, that p27 may stabilize binding of these TFs to these sites, since it presumably lacks DNA binding ability (which STAT3 and cJUN clearly have).

2. The authors use the word "unprecedented" two times in the manuscript. In neither case is the use appropriate (that is, uncovering a relationship that has never been shown before).

Response: The word "unprecedented" has been removed from two places in the manuscript.

Reviewer #2 In this manuscript, the authors study the kinase-independent role of p27 c-terminal phosphorylation in regulating breast cancer stem cells and cancer progression. While kinase independent roles of p27 have been reported in cancer progression, studying the involvement in cancer stem cells has novelty. They found strong evidence that p27pT157pT198 could regulate mammary stem cells and cancer progression using both cell culture and in vivo limiting dilution transplantation assays. They also discovered a novel mechanistic underpinning for such a role of p27 involving STAT3 and a few other transcriptional co-regulators. However, there are a number of concerns especially in lacking knockin confirmation.

Main concerns:

1. The major conclusion that cancer stem cell (CSC) abundance is transcriptionally regulated by C-terminally phosphorylated p27 (p27pT157pT198) is based on experiments using overexpression of a p27 mutant that is kinase dead and has both T157 and T198 mutated to a phosphomimetic residue (D). But

GEORGETOWN UNIVERSITY MEDICAL CENTER

LOMBARDI COMPREHENSIVE CANCER CENTER
Research • Education • Treatment

this overexpression strategy is not adequate to prove the main conclusion. The authors should confirm their major findings using crispr knock-in of this phosphomimetic. Furthermore, phosphomimetic gain of function mutants should be accompanied by loss of function mutants (such as alanine mutants that prevents phosphorylation) which should also be knocked into the genome.

Response: Our major conclusion that p27 modulates transcription by STAT3 and cJun to activate programs of stem cell self-renewal is supported not only by expression of p27CK-DD in 231, UMUC3, MCF7 and MCF-12A (and now of 27CK-AA), but also by the consequences of knockdown of endogenous cellular p27 in the 1833 and UMUC-Lu3 lines with constitutive PI3K activity and high endogenous p27pT157pT198. It is also supported by the mouse model in which TG p27CK-DD is expressed in mice in which bi-allelic endogenous murine p27 remains present, indicating that the phosphomimetic p27 has a dominant active phenotype to mediate hyperplasia and yield microinvasive cancers some of which metastasize.

As requested, effects of p27CK-DD expression are now complemented with new assays in which p27CK-AA is expressed. p27CK-AA fails to upregulate ES-Ts (Fig 1C top) and tumor spheres (Fig 1C bottom), and ANGPTL4 and JAG1 gene expression (Fig 4A, Fig S3A), and does not repress PTPN12 expression (Fig 5B). In these models, p27CK-DD is not overexpressed. It is expressed at levels similar to those in cancer lines with constitutive PI3K activity and detectable p27pT198pT157.

p27 is a small unstructured protein with no known catalytic function that binds and inhibits cyclin CDKs with 1:1 stoichiometry. It also carries out non-CDK dependent actions by binding TFs and by binding RhoA /ROCK1 to alter the actomyosin cytoskeleton. The CK- refers to 4 amino acid changes that abolish binding to both cyclin and CDKs, preventing cyclin CDK inhibition by p27 (Vlach EMBO 1997). Use of the CK- mutations allow us to separate the CDK inhibitory function from the role of C-terminal phosphorylation of p27 in transcription and to test the effect of inserting phosphomimetic mutations at T157 and T198. This paper does not address the potential role of p27 bound to cyclin/CDK2 in transcriptional activity; this is under investigation, but it is a significant project on its own and will not be included here.

A large body of work published over the last 2 decades indicates that p27DD and /or p27CK-DD have significant effects on p27 localization, stability, protein/protein interactions and promote tumor cell migration, invasion, metastasis using assays that did not knock out endogenous p27 (see refs #23-25, 33, 34, 37, 38). We also previously showed p27pT157pT198 and p27CK-DD coregulate cJun to induce transcription of cJun target TGFB2, and other gene drivers of tumor metastasis (Yoon et al, PNAS 2019). These papers all used cells with p27CK-DD levels similar to endogenous p27 in cancer lines with constitutive PI3K activation. Even in cells with high endogenous p27pT157pT198, non-phosphorylated p27 is detectable, so the expression of transduced phosphomimetic p27 in the presence of endogenous p27 mimics the coexistence of non-phosphorylated p27 and p27pT157pT198 in cells. That p27CK-DD can have significant effects despite persistence of endogenous p27 is also shown by the mammary expression of TG p27CK-DD in the MMTVCre X p27CK-DD bigenic model presented herein.

In summary, this paper shows effects of cellular p27 in PI3K activated breast (1833) and bladder (UMUC3-Lu2) cancer lines and of p27CK-DD in 231, MCF7, MCF12A and UMUC3 lines. While the CRISPR models are elegant, their inclusion herein would very significantly delay publication of this work and we do not think CRISPR KO of WT p27 and KI of the two different phosphomutants p27CK-DD and p27CK-AA

GEORGETOWN UNIVERSITY MEDICAL CENTER

LOMBARDI COMPREHENSIVE CANCER CENTER
Research • Education • Treatment

will substantially strengthen our conclusions regarding action of C-terminally phosphorylated p27 in the presence of WT p27.

2. The authors soften many of their conclusions at the end of experiments or even a series of experiments by using words such as “appear,” leading readers to question how confident the authors are in their experimental design and execution. In particular, this reviewer questions the validity of their section title “p27 phosphorylation increases ES-TF expression and sphere forming cell abundance” when the final conclusion statement of this section is “Taken together, these data indicate that p27pTpT or p27CK-DD expression appears to increase stem-like cells and stem cell-related gene expression.

Response: We agree with the reviewer that the word “appear “ was overused. We have removed the word “appear”, changing the wording to reflect our confidence in the data, while retaining appropriate scientific caution. See tracked changes.

3. The authors state that the “STAT3 binding peaks were significantly reduced upon p27 depletion (Figure 2A right)”, but there is no statistical analysis attached to this statistical term and the cited van diagram shows a very modest reduction (from 38k to 33k).

Response: This section in the results has been rewritten for greater clarity on tracked version, page 8. Venn Diagram Fig 2A left shows that over 16K of 38K STAT3 binding sites are also sites of p27 binding. We now state that “STAT3 binding to p27-STAT3 co-bound sites was significantly reduced upon p27 depletion (Fig 2A right, and 2C).The heat map Fig 2C middle portion shows that of 16,697 STAT3 peaks co-bound by p27, 73% of these or 12,248 were lost or decreased in amplitude with p27 knockdown.” Furthermore, Fig 2F (right) shows that the amplitude of binding of STAT3 at p27/STAT3 co-target gene sites was significantly reduced by p27 knockdown. The adjusted p value of 2.81E-04 is now included in Fig 2F which displays p27-STAT3 target genes, reflecting its high statistical significance.

4. The authors state that “DNA binding motif analysis showed that both STAT3 and Jun/AP1 motifs are frequently found within p27, STAT3 individually bound sites and at p27/STAT3 co-occupied sites (Figure 2H),” but the cited Fig 2H does not appear to show all of the data referenced. Also this panel’s lettering is a bit confusing.

Response: The data for STAT3 and Jun /AP1 motifs located at sites of p27, STAT3 or sites bound by p27/STAT3/cJun are now all included in Fig 2H. The figure has been revised to show the frequency (% of targets and adj p value) of STAT3 and Jun/AP1 motifs at all p27 binding sites, all STAT3 binding sites and at all sites co-bound by p27/STAT3 and reworded for greater clarity-see tracked version, page 8 bottom.

5. The cutoff for high expression in KMP in Fig 3 (1212 for high expression and 2439 for low expression) lacks justification. The same is true for other KMP of human data.

Response: The KM Plotter software identifies the cut offs for low and high gene automatically by as the expression levels that separate good and bad outcome groups with the greatest statistical significance. The cut offs were not predetermined by the researchers. This is clarified in the Methods section

GEORGETOWN UNIVERSITY MEDICAL CENTER

LOMBARDI COMPREHENSIVE CANCER CENTER
Research • Education • Treatment

6. While it is possible that p27 binds to target genes first and then recruits STAT3 and c-Jun, the data are also consistent with that STAT3 binds first and then recruits p27. So caution should be exercised in interpretation.

Response: We agree. p27 might be recruited after the TFs and we have changed wording to reflect this.

7. The use of PI3K inhibitor (PF1502) to link c-terminal phosphorylation of P27 to activation of Pyk2 and Stat3 activation needs to be very cautious. PI3K likely can regulate other substrates that can also affect p27 activity.

Response: We showed in Nature Medicine 2002, PNAS 2006 and Molecular Cell 2008 that p27 is phosphorylated at T157 and T198 by AKT, RSK and SGK, respectively. Likely other AGC kinases do so also in a context dependent manner. We agree that the PI3K inhibitor would affect these also and might have other substrates that modify p27 in other ways. Fig 5 shows that 231DD and 1833 have reduced levels of PTPN12 the phosphatase that dephosphorylates and inactivates Pyk2. Pyk2 is known to catalytically activate STAT3. 231DD and 1833 cells with low PTPN12 also have increased Pyk2 and STAT3 activation (Fig 5B,C). Fig 5D shows PI3K inhibition (PF1502) inhibits accumulation of p27pT198, and leads also to loss of pPyk2 without loss of total Pyk2. Further, loss of PYK2 by siRNA or its inhibition by PF431396 reduces activated phosphoSTAT3 but does not affect p27pT198 levels. These data are consistent with PI3K dependent p27pT198 acting to repress PTPN12, with resulting activation of Pyk2 and its activation of STAT3.

8. Justifications should be given to why different subsets of stem cell factors (NANOG, SOX2, OCT4 vs. NANOG, and SOX2, and cMYC,) are assayed for knockdown vs. overexpression and between different cell lines. It is unclear why not all four factors are measured for all treatments to be consistent.

Response: NANOG, SOX2, OCT4, KLF4 and cMYC were all measured in Fig 1 and FigS1 and other Figs. Of the ES-TFs shown in data herein, all were tested in all lines with and without DD transduction or shp27 and in the assays testing loss or inhibition of STAT3. We now indicate this in the methods section and state that negative data were not graphed. It is not clear why STAT3 loss through STAT3CR and siRNA in different lines reduce Nanog and Oct 4 to different degrees in these assays. While we do not have a good explanation for this, rather than leaving out or trimming the data, we show results as obtained.

9. The conclusion that "Cancers were more prevalent, and more abundant within the MG from p27CK-DD mammary TG mice (n=9/11 with mammary cancers) (Figure 7C,D)" is problematic. The cited panel C shows small precancerous lesions that are not yet tumors. Only palpable masses should be considered to be tumors and Kaplan-Meier tumor free plots should be presented in Fig 7 to better summarize the tumor data.

Response: The results section for Figure 7 has been re-written for greater clarity. The nomenclature has been clarified. A neoplasm does not need to be palpable to be called cancer. A breast cancer is an invasive mammary epithelial neoplasm that extends beyond the mammary ducts. We have changed the word "cancer" to "microinvasive cancer". The microfoci of cancer in these mammary glands were invasive, with irregular borders that clearly invaded beyond the basement membranes into fat and in all of the glands showing these lesions. These glands contained both hyperplastic lesions as well as frank malignant microinvasive cancers. All of the tumors were detected at the 18 months at necropsy, so a KM

GEORGETOWN UNIVERSITY MEDICAL CENTER

LOMBARDI COMPREHENSIVE CANCER CENTER
Research • Education • Treatment

tumor free plot would not depict this best. Instead the number of mice that developed tumors is graphed for the 3 genotypes (Fig 7c and table Fig. S6c). In p27CK-DD TG mice, more mice showed the presence of these lesions than in p27CK- TG or MMTV Cre controls (n=12/14 for p27CK-DD, 6/14 for p27CK- and 2/14 for MMTV Cre controls). The microinvasive cancers were also more numerous within each affected gland evaluated among the p27CK-DD TG mice than the other 2 genotypes- which is appreciable on the whole mount low power view. In 6/12 p27CK-DD TG mice with these mammary micro tumor foci, these gave rise to liver metastasis. Only the largest 4th mammary fat pads were evaluated histologically, so larger primary cancers might have been present in other glands. None were palpable. We have explained this more clearly in the revised results section.

10. The Mouse 26 panel in Fig 7 appears to a lymph node rather than mammary lesions.
Response: Mouse 26 showed an invasive cancer of a little over 2 mm. We agree, this tumor involved and invaded beyond an intramammary lymph node. We now provide higher magnification and higher resolution images and cytkeratin staining that reveal the cancer cells invading into peritumoral fat surrounded by abundant lymphocytes. All histopathology was reviewed by our collaborating internationally known breast cancer Pathologist Dr Tan Ince.

11. Higher resolution photos in Fig 7 are needed to substantiate the claim for “strong staining for the transgenic p27CK-pT198 protein in both cytoplasm and nucleus”
Response:Higher resolution images provided.

12. Please ensure that liver mets in Fig 7 are real. The current images appear to show inflammatory cells in liver.
Response: We agree it is critical to firmly establish that these liver lesions are metastatic. All 3 of the initial p27CK-DD with liver micrometastases were PCR+ for TG p27CK-DD, while Cre only control livers (n=5) were negative by PCR for the TG. High resolution micrographs of liver lesions show epithelial clusters with associated lymphocytes Fig 7 and Fig S6e. It was not possible to restrain these slides for cytkeratin and transgene protein using p27pT198. Because those liver blocks were lost by Fed Ex during my lab’s move from Miami to DC during COVID, we have now evaluated mammary and liver tissues in an additional 3 p27CK-DD mice, and found that both histopathologically and by PCR, all contained both microscopic intramammary invasive cancers and hepatic micrometastases. A representative liver met in revised Fig 7 stained for p27pT198 and cytkeratin, absent in the surrounding hepatic tissues. Thus, 6/12 p27CK-DD mice with intramammary neoplasms showed liver micrometastasis as confirmed histologically and by PCR for the transgene. These were reviewed by our collaborating Pathologist (Dr Tan Ince) and two murine mammary cancer expert co-authors (M Johnson and K Briegel) who concur that they represent epithelial cancer metastasis histologically.

13. The authors stated that qPCR detected transgene expression in liver without showing the data. Please show the data as part of supplementary figs and ensure proper controls to ensure that the signal is due to metastasis but not leaking expression of the transgene. A good negative control is the liver before primary tumor detection.

GEORGETOWN UNIVERSITY MEDICAL CENTER

LOMBARDI COMPREHENSIVE CANCER CENTER
Research • Education • Treatment

Response: Expression of MMTVA promoter driven Cre is limited to the mammary and salivary glands. MMTV Cre does not drive TG expression in the liver. Thus, the presence of the TGp27CK-DD in the liver reflects metastasis of mammary cancer to the liver. Liver tissue RNA extracted from p27CK-DD and Cre only control mice confirms TG presence in the liver in all of 6 affected p27CK-DD livers but none of 5 control livers (Data now shown in Fig S6f, PCR used 2 different primer sets- one for p27TG and a primer set that spans tdTomato and p27).

Responses to Minor Concerns

1. Please state in Results whether the p27 and STAT3 CHIP data were generated by the authors for this study or previously published.

Response: We state again and more clearly in Results and Methods that the p27 and Jun ChIP seq data were previously published in PNAS 2019. The STAT3 CHIP data were generated by the authors for this study.

2. The statement “p27 might transcriptionally co-activate STAT3 to drive genes governing CSCs expansion” is confusing and may be interpreted to suggest that p27 is involved in transcription of STAT3.

Response: Confusing statement revised: “p27 co-activates STAT3 to promote the transcription of genes governing CSC expansion”.

3. Fig 4C is cited for “transduction of p27CK-DD into 231 increases activated phospho-isoforms of both Pyk2 and STAT3 (Figure 4C).” But the actual data appears to be in Fig 5C.

Response: Correction made: mislabeled Fig 4 is changed to Fig 5C.

4. There is a punctuation error in the following sentence: “cMyc, is an embryonic stem cell transcription factor whose dysregulation mediates malignant cell proliferation, immortalization, angiogenesis, metastasis and CSC self-renewal.”

Response: The comma after Myc was removed from the sentence.

Reviewer #3, Razavipour and coworkers describe in this interesting manuscript a novel CDK-inhibitory independent function of the CDK inhibitor p27Kip1 in transcriptionally regulating genes involved in stem cell expansion and self-renewal, leading to cancer stem cell expansion. p27 becomes recruited in complex with STAT3/CBP to genes including MYC, ANGPL4 or JAG1. It also recruits HDAC1/SIN3A complexes to the PTPN12 gene, which leads to activation of Pyk2. To demonstrate physiological significance, the authors a transgenic mouse model of mammary expression of a CDK/cyclin-binding defective C-terminal phosphomimetic p27, that causes increased mammary duct branching, hyperplasia and invasive metastatic cancer.

These data expand previous reports from the Slingerland lab, which recently used this model to show that C-terminally phosphorylated p27 is a coactivator of cJun and drives tumour progression and metastasis. They also demonstrated that cytoplasmic p27CK-DD binds STAT3, induces Twist1 and promotes EMT.

The present manuscript adds important novel insights in the CDK-independent transcriptional functions of C-terminal phosphorylated p27. One of the main conclusions, that PI3k-signalling alters the transcriptional function of p27 by C-terminal phosphorylation should be tested more directly, by comparing the phosphomimetic mutant p27CK-DD to a nonphosphorylatable mutant p27CK-AA. The mutant CK-AA should also be included in IPs to exclude effects of CDK/cyclin binding/recruitment

GEORGETOWN UNIVERSITY MEDICAL CENTER

LOMBARDI COMPREHENSIVE CANCER CENTER
Research • Education • Treatment

deficiency of CK-DD on DNA. The cell lines 231 and 1833 might not be ideal for this since the ratio (pT198 p27 vs. total p27) is rather similar in 231 and 1833 cells (Fig. 5D). Prior to publication in Nature Communications, the missing controls discussed below should be included for key experiments.

Main Points:

1. Fig. 1A,B: This figure is very similar to a figure shown in the previous paper (Yoon et al., 2019). The authors might comment on the novel information (more regulated genes have been identified) and why some of the previously top scoring pathways (e.g. ECM receptor interaction; Protein digestion and absorption) were not identified in the novel pathway analysis.

Response: Fig. 1A shows an analysis similar to that in Yoon et al., 2019 but the fold change for induction was 1.5 instead of 2 in Yoon et al. In addition, downregulated genes were not displayed in Venn diagrams in Yoon et al. For the Pathway analysis, we used updated Wiki 2021 (not 2016 as in Yoon et al) leading to changes in pathways identified. The Venn Diagrams and the pathway analysis of p27-downregulated genes are novel. This is made clearer in the Result and Methods section

2. Fig. 1C: Does expression of EGFP-p27CK-AA to level as EGFP-p27CK-DD lead to no increase of SOX2, NANOG or cMYC? Is there no increase in sphere formation?

Response: As requested, we now show p27CK-AA transduction in 231 led to no increase in spheres and ES-TF (MYC, SOX2 and NANOG) expression. As expected, the levels of expression of p27CK-AA in 231 cells was lower than that of p27CK-DD due to its shorter T1/2 (Malek 1999, see new Fig S1A). p27CK-AA fails to increase ES-TFs, ANGPTL, tumor spheres, and does not repress PTPN12 expression.

3. Fig. 1E,F: T198 phosphorylation increases its stability. Are protein level of p27CK- and p27CK-DD similar? Is p27CK- not phosphorylated on T157 and T198? What is the CD44 expression of CK-?

Response: In most cell types, p27CK- levels are about half those of p27CK-DD due the stabilizing effect of the phosphomimetic, DD mutations (shown for the 231 line in Fig S1A). The p27CK- can be phosphorylated by endogenous PI3K effector kinases (AKT, SGK, RSK) but these are not highly activated in the immortal but not malignantly transformed mammary MCF12A line.

As requested we now show CD44 levels are higher in p27CK-DD than p27CK- expressing MCF12A (Fig 1F, right). The effects of p27CK-DD vs p27CK- on CD44 levels and sphere numbers exceed the differences in steady state levels of the proteins in MCF12A cells.

4. Fig. 1G: There is a scale bar, but its size is missing. How many cells are represented in the “dots” in vector or CK-? Could the colony number of CK- be included in the bar graph?

Response: The Scale bar has been restored to Fig 1G. Colony numbers for CK- graphed in Fig 1G are now presented clearly. It is really not possible to estimate the number of cells in these tiny vector and CK- colonies. A 100 uM colony might have 100-120 cells.

5. Fig. 2C: Does the p27 DNA-binding heat map change upon PI3k inhibition? Or is DNA binding of p27 not altered upon C-terminal phosphorylation?

Response: We did not test the effects of PI3K inhibition on global p27 recruitment to DNA and a repeat of the entire CHIPseq with PI3K inhibitor is beyond the scope of this revision. To address this concern, for the

GEORGETOWN UNIVERSITY MEDICAL CENTER

LOMBARDI COMPREHENSIVE CANCER CENTER
Research • Education • Treatment

ChIP-PCR validation of target genes, we now provide data with and without PI3K inhibitor, PF1502, for ChIP-PCR of p27, STAT3, cJun, CBP, H3K27Ac, SIN3A, YY1, and HDAC1 on MYC (Fig 3), JAG1 (Fig S3), ANGPTL4 (Fig 4) and PTPN12 (Fig 5). As previously published (Larrea PNAS 2009, Wander BCRT 2013) and shown in Fig 5D, short term PI3K inhibition over 48 hrs decreases p27pT198 on WB, without decreasing total p27 levels. These new data show p27 recruitment to DNA at these targets is reduced by PI3K inhibition, indicating that p27 phosphorylation promotes its binding to these targets.

6. Fig. 3F: The occupancy of the MYC promotor by p27 does not significantly change between 231 and 231DD. Does this mean that binding is independent of C-terminal phosphorylation? Or is only phosphorylated p27 bound? Also here, PI3K inhibition might change binding. Why is binding of STAT3 generally not shown for 231 and 231DD?

Response: The ENCODE tracks for p27 binding to the MYC promoter (Fig 3F) actually show greater amplitude in 231DD and 1833 than in 231. This is also validated in the ChIP PCR data presented in Fig 3H that shows p27CK-DD transduction increased p27-DNA binding on ChIP-PCR at the MYC promoter site at -1300 bp as compared to 231. New ChIP PCR data in this revision also shows reduced p27 recruitment to this MYC site with prior treatment with PI3K inhibitor, PF1502 (Fig 3H). ChIPseq for STAT3 was carried out in 1833 and 1833shp27 only, so it is not shown for 231 and 231DD. This is indicated in the Results and Methods sections.

7. Fig. 3H: Does the increase in 231DD reflect the degree of p27 overexpression? A control 231-AA is missing here. This control would also be important for Fig. 3J-L and other p27 ChIP (e.g. Fig. 5F). Without this control, the conclusion on page 9 "Thus p27pTpT appears to induce genes ... by enhancing stable STAT3 and cJun recruitment" might be an overinterpretation.

Response: The expression of p27 in 231p27CK-DD is about 2 fold that compared to control endogenous p27 in 231 (see Fig S1A). Please note that the increase of p27 CK-DD binding to the MYC promoter in Fig 3H considerably more than 2-fold higher than that in 231. Since p27CK-AA transduction into 231 failed to induce MYC vs 231 controls (now shown in Fig 1C top), we did not include it in the ChIP assay in Fig 3s. Moreover, since the p27CK-AA is rapidly turned over (Malek EMBO 2009, Liang NCB 2004), its inclusion in the ChIP assays would be hard to interpret. These data support our conclusion. We show less binding of p27 to target gene promoters in 231 than in 1833 whose endogenous p27pT198 levels differ, and phosphomimetic p27CK-DD shows greater binding than WT p27.

8. Fig. 3M, Fig. 4C: What antibody was used for the p27 IP? Could it be that the antibody does not precipitate CDK-bound wt p27? It appears that EGFP-p27 is selectively precipitated.

A control of EGFP-p27AA is missing. Without this control, it cannot be judged if p27-DD enhances complex formation. It would be a concern if endogenous wt p27 does not bind endogenous cJun, CBP or STAT3. According to the model of the authors, p27 of 1833 cells should precipitate comparable level of cJun, CBP or STAT3. Has that been tested?

Response: The antibody used in Fig 3M is the BD p27 monoclonal that does precipitates endogenous p27. Cellular p27 runs faster than the larger GFP-tagged p27CK-DD, but is clearly present in the p27 IP in the 231 lane. Because p27CK-AA is unstable, the experiment proposed will not support or refute that p27DD enhances complex formation. Fig 3M shows that endogenous p27 in 231 binds STAT3 and cJun in 231

cells. The complex between endogenous p27 and cJun was also shown by co-IP and by proximity ligation assay in Yoon 2019.

According to the model of the authors, p27 of 1833 cells should precipitate comparable levels of cJun, CBP or STAT3 as in 231DD. Has that been tested?

Revised Fig 3M now includes a p27IP from 1833. This shows that showing p27 bound cJun and Stat3 in 1833 and 231DD are both higher than in 231.

9. Fig. 5B: 231-AA would be an important control.

Response. We now show that PTPN12 is not repressed in 231CK-AA line. Having shown this, addition of 231p27CK-AA to all of the CHIP and IP data is not required for the conclusions drawn from the data (See points 11-13 below).

10. Fig. 5G: In this STAT3 CHIP, the signal in 231DD seems comparable to 231 and 1833shp27 – and much lower than in 1833. Any explanation why?

Response: The increase in STAT3 binding to the PTPN12 site in 231DD versus 231 is statistically significant. These data represent the mean of triplicate repeat CHIP-PCR in each of 3 separate biologic assays.

11. Fig. 5L: The control 231-AA is missing here

See response to 9 above

In addition to the phosphomimetic mutations, EGFP-p27CK-DD could also act different because it fails to bind CDK/cyclins or the EGFP portion might alter the complex composition.

Response: The binding to STA3 and Jun and transcriptional effects of p27CK-DD are reproduced by cellular p27 in the 1833 line which binds readily to cyclin CDKs, indicating that the DNA binding effects of p27CK-DD are due to its phosphomimetic effects and not due to failure to bind cyclin/CDKs. The GFP tag is N terminal and does not appear to interfere with the effects of C-terminal modifications required for p27 chromatin binding since p27 binding profiles in 231DD and 1833 were similar (Yoon 2019). The extent of p27 binding to DNA in 231CK-DD and 1833 are similar as published in Fig 6 I from Yoon PNAS 2019, shown here).

12. Fig. 6G. shPyk2 reduced ALDH1 activity in 231DD far below levels of 231. Does this mean that shPyk2 might act independent on ALDH1 activity?

Response: We agree this likely indicates that Pyk2 has p27 independent effects on cell cycle. This could reflect additional effects on Src and STAT3 activation. This is mentioned in the revised discussion pg 18.

13. Fig. 7: T298 phosphorylated p27 has been reported to be much more stable than p27. What are the protein expression level of CK- and CK-DD?

GEORGETOWN UNIVERSITY MEDICAL CENTER

LOMBARDI COMPREHENSIVE CANCER CENTER
Research • Education • Treatment

Response: The relative amounts of p27 and p27pT198 cannot be compared based on the difference in intensity of two different p27 antibodies on IHC. The relative affinities of the total p27 Ab from BD and of the Thermo p27pT197 antibody for their targets on IHC are not known, making direct comparison inappropriate. We do not have fresh tumors to IP total p27 to compare the GFP-p27 levels in these tumors- it would take well in excess of 24 months to re-do the breeding and carry the mice to 18 mo to do this.

Minor points:

Response: Thanks for your careful editing. All legends have been corrected.

Suppl. Fig. 1G: the labelling of the STAT3 western blot must be swapped.

Response: Suppl. Fig. 1G: the labelling of the STAT3 western blot has now been swapped.

The figure legend for Suppl. Fig. 1 should be corrected ((G-H shown no colony forming assay...)).

Response: The figure legend for Suppl. Fig. 1 has been corrected to remove reference to the colony forming assay.

Legend for Fig. 6K (table T-ISC frequency) is missing. The legend for Fig. 6K is for Fig. 6L (heatmap).

Response: Legend for Fig. 6K (table T-ISC frequency) has been added. Fig. 6L heatmap legend corrected.

Legend for Figure S5 H,I,J: J is included in H; H and I are swapped.

Response: Legend for Figure S5 H,I,J: J has been removed and H and I reversed.

Legend for Figure S6 is missing.

Response: Legend for Figure S6 has been added.

Reviewers' Comments:

Reviewer #1:

Remarks to the Author:

The authors have adequately addressed the substantive issues that had been raised.

Reviewer #2:

Remarks to the Author:

The authors have largely addressed my previous concerns and the manuscript has strengthened, but there are a few remaining concerns.

1. The authors now provide a reasonable explanation for why Kaplan-Meier tumor free survival plots cannot be provided. Yes, invasive tumors can be microscopic, but quantifying these small cancers requires a bit more than H&E. IHC data for myoepithelial markers such as p63 are needed to confirm invasiveness of these microscopic lesions. In addition, besides percentage of cancer-bearing mice, tumor multiplicity, which provides a more sensitive method for comparing these genotypes, should be plotted.

2. Selecting different subsets of the five stem cell markers to present in different figures causes confusion. This reviewer strongly suggests that all five markers are shown in all figures regardless of whether the data are positive or negative.

Reviewer #3:

Remarks to the Author:

The authors have addressed and/or discussed all the points raised in my review. In their revised manuscript, the authors now include additional data that strengthen the manuscript and support the main conclusion.

In response to some suggestions to test the mechanism more directly (e.g. my previous points #7 or #8 ...), the authors emphasise that the enhanced stability of p27-CK-DD will complicate a comparison with p27-CK-AA and make the results difficult to interpret. This was also my concern, but combining a stabilising mutation (e.g. T187A) with p27-CK-AA to increase the stability of the unstable mutant might help to overcome this problem - but I agree that there is no guarantee that this approach would work.

The manuscript adds important new evidence for a CDK-independent transcriptional function of p27 in cancer stem cell regulation that is of general interest for the cell cycle and cancer fields. I recommend publication in Nature Communications.

GEORGETOWN UNIVERSITY MEDICAL CENTER

LOMBARDI COMPREHENSIVE CANCER CENTER
Research • Education • Treatment

April 23, 2024

Responses to Reviewers

Please see responses to reviewers below.

The reviewer comments are in regular font and our responses are italicized.

Reviewer #1 (Remarks to the Author):The authors have adequately addressed the substantive issues that had been raised.

Response: Reviewer 1 have accepted the paper without further revision.

Reviewer #2 (Remarks to the Author):The authors have largely addressed my previous concerns and the manuscript has strengthened, but there are a few remaining concerns.1. The authors now provide a reasonable explanation for why Kaplan-Meier tumor free survival plots cannot be provided. Yes, invasive tumors can be microscopic, but quantifying these small cancers requires a bit more than H&E. IHC data for myoepithelial markers such as p63 are needed to confirm invasiveness of these microscopic lesions. In addition, besides percentage of cancer-bearing mice, tumor multiplicity, which provides a more sensitive method for comparing these genotypes, should be plotted.2. Selecting different subsets of the five stem cell markers to present in different figures causes confusion. This reviewer strongly suggests that all five markers are shown in all figures regardless of whether the data are positive or negative.

Response: We have now carried out p63 staining of the microinvasive cancers and provide representative data for p27CK-DD X Cre bigenic mammary gland from mouse 1668. As can be seen in the fig S6, p63 stained myoepithelial cells surround the normal ducts circumferentially. In the same MG tumor shown in figure 7d top, the circumferential stain is lost and the tumor cells are invading the fat. Notably, the hyperplastic gland at the top right of fig 7d shows myoepithelial cells that do not fully surround the gland. Tumor multiplicity has been provided.

We respectfully disagree with the reviewer about showing all of the negative data for individual stem cell transcription factors. We state clearly in the manuscript that all were tested for all of the lines and conditions, and the negative data were not shown.

Reviewer #3 (Remarks to the Author):The authors have addressed and/or discussed all the points raised in my review. In their revised manuscript, the authors now include additional data that strengthen the manuscript and support the main conclusion. In response to some suggestions to test the mechanism more directly (e.g. my previous points #7 or #8 ...), the authors emphasize that the enhanced stability of p27-CK-DD will complicate a comparison with p27-CK-AA and make the results difficult to interpret. This was also my concern, but combining a stabilising mutation (e.g. T187A) with p27-CK-AA to increase the stability of the unstable mutant might help to overcome this problem - but I agree that there is no

GEORGETOWN UNIVERSITY MEDICAL CENTER

LOMBARDI COMPREHENSIVE CANCER CENTER
Research • Education • Treatment

guaranteethat this approach would work.The manuscript adds important new evidence for a CDK-independent transcriptional function of p27 in cancer stem cellregulation that is of general interest for the cell cycle and cancer fields. I recommend publication in NatureCommunications.

Response: Reviewer 3 have accepted the paper without further revision.